# Covariate-dependent Graphical Model Estimation via Neural Networks with Statistical Guarantees

**Jiahe Lin** [⋆]
*Machine Learning Research, Morgan Stanley*

**Yikai Zhang** [⋆]
*Machine Learning Research, Morgan Stanley*

**George Michailidis** ✉
*Department of Statistics and Data Science*
*University of California, Los Angeles*

**Reviewed on OpenReview:** *https://openreview.net/forum?id=beqSqPgE33*

## Abstract

Graphical models are widely used in diverse application domains to model the conditional dependencies amongst a collection of random variables. In this paper, we consider settings where the graph structure is covariate-dependent, and investigate a deep neural network-based approach to estimate it. The method allows for flexible functional dependency on the covariate, and fits the data reasonably well in the absence of a Gaussianity assumption. Theoretical results with PAC guarantees are established for the method, under assumptions commonly used in an Empirical Risk Minimization framework. The performance of the proposed method is evaluated on several synthetic data settings and benchmarked against existing approaches. The method is further illustrated on real datasets involving data from neuroscience and finance, respectively, and produces interpretable results.

## 1 Introduction

An undirected graphical model captures conditional dependencies amongst a collection of random variables and has been widely used in diverse application areas such as bioinformatics and the social sciences. It is associated with an undirected graph $G = (V, E)$ with node set $V := \{1, \cdots, p\}$ and edge set $E \subseteq V \times V$ that does not contain self-loops, i.e., $(j, j) \notin E; \forall j \in V$; $\boldsymbol{x} = (x_1, x_2, \cdots, x_p)'$ is a random real vector indexed by the nodes of $V$, with probability distribution $\mathbb{P}$.

The connection between the graph $G$ and the probability distribution $\mathbb{P}(\boldsymbol{x})$ comes through the concept of graph factorization and the properties that $G$ exhibits. The probability distribution $\mathbb{P}(\boldsymbol{x})$ *factorizes* with respect to $G$, if it can be written as $\mathbb{P}(\boldsymbol{x}) = \frac{1}{Z} \prod_{C \in \mathcal{C}(G)} \varphi_C(\boldsymbol{x}_C)$, where $\mathcal{C}(G)$ denotes the set of all cliques of $G$, $\varphi_C > 0, \forall C \in \mathcal{C}(G)$ potential functions, and $Z$ a normalizing constant. If the underlying graph $G$ in addition exhibits a Markov type property (see, e.g., Lauritzen (1996) and Appendix D.3), then the conditional dependence relationships are *encoded* in the potential functions $\varphi_C$. In particular, for $G$ satisfying the *pairwise Markov* property, $\mathbb{P}$ can be written as $\mathbb{P}(\boldsymbol{x}) \propto \prod_{\{j,k\} \in E} \varphi_{\{j,k\}}(\boldsymbol{x}_{\{j,k\}})$, and the absence of an edge in $G$ implies that the corresponding potential function is zero and consequently the nodes are conditional independent given the remaining nodes. The upshot is that the conditional independence relationships for $\boldsymbol{x} \sim \mathbb{P}$ can be read-off from the graph, if $\mathbb{P}$ factorizes w.r.t. $G$ and the latter is Markov.

For certain multivariate distributions, the form of the joint distribution $\mathbb{P}(\boldsymbol{x})$ enables its decomposition into clique-wise potential functions based on $G$ in a fairly straightforward manner; examples include the Gaussian

---

[⋆] Equal contribution, ordered alphabetically. ✉ Corresponding Author. E-mail: `gmichail@ucla.edu`.

for continuous variables (Lauritzen, 1996, see also Appendix D.1), and the Ising and the Boltzmann machine models (Wainwright et al., 2008) for binary and discrete random variables, respectively. The corresponding potential functions $\varphi_C(\cdot)$ are parameterized by (functions) of the parameters of the underlying distribution (see example in Appendix D.1). Consequently, a graphical model and the conditional independence relationships that $G$ encodes can be obtained by estimating the parameters of the underlying distribution, either through maximum likelihood (Banerjee et al., 2006; Friedman et al., 2008) or regression procedures using neighborhood selection techniques (Meinshausen & Bühlmann, 2006).

For other classes of graphical models beyond those mentioned above, to select the clique-wise potential functions $\varphi_C(\cdot)$ becomes more involved. The difficulty stems from specifying potential functions based on which a compatible joint distribution $\mathbb{P}(\boldsymbol{x})$ with respect to $G$ can be obtained. One popular approach is to define node conditional distributions through pairwise cliques $\{j, k\}$; i.e., for *any* fixed $j$, $\mathbb{P}(x_j|\boldsymbol{x}_{-j}) \propto \prod_{k \in \text{ne}(j)} \varphi_{\{j,k\}}(x_j, x_k)$ for $k$ in the underline{neighborhood} of $j$. Such a conditional distribution captures *pairwise* conditional interactions between nodes, however, this construction does not directly define a proper graphical model for the joint distribution $\mathbb{P}(\boldsymbol{x})$, unless additional conditions are satisfied (see example in Appendix D.2). In general, the question of the compatibility of a graphical model for the joint distribution $\mathbb{P}(\boldsymbol{x})$ based on a particular specification of all the node conditional distributions in $V$ is rather open, and is generally addressed on a case by case basis (Berti et al., 2014). In this paper, we adopt the convention that models specified through their node-conditional distributions are also referred to as graphical models, which capture node-wise conditional independence relationships.

The presentation thus far focused on a single "static" graphical model, which is suitable for modeling tasks involving *homogeneous* data sets; i.e., the observed samples $\boldsymbol{x}^i, i = 1, \cdots, n$ are iid from $\mathbb{P}(\boldsymbol{x})$. However, in certain applications, the conditional independence relationships may be *heterogeneous* in that they are modulated by external covariates $\boldsymbol{z} \in \mathbb{R}^q$. An example from biology is that the brain connectivity network—wherein the coordinates of $\boldsymbol{x}$ correspond to different brain regions—varies with age ($\boldsymbol{z} \in \mathbb{R}_+$). Similarly, gene expression exhibits individual-level variation affected by single nucleotide polymorphisms (SNPs); see, e.g., Kim et al. (2012). The SNPs are thereby covariates—potentially of large dimension—that impact the corresponding gene co-expression network. Motivated by such applications, this work introduces a class of *covariate dependent, pairwise interaction* graphical models, wherein the *log-densities* of the node conditional distributions are defined by

$$x_j = \sum_{k \neq j}^p \beta_{jk}(\boldsymbol{z})x_k + \varepsilon_j, \quad j \in \{1, \cdots, p\}, \tag{1}$$

where $\boldsymbol{z}$ is a $q-$dimensional observed covariate, $\beta_{jk}(\cdot) : \mathbb{R}^q \mapsto \mathbb{R}$ functional regression coefficient dependent on $\boldsymbol{z}$ that impacts the strength of the interaction between nodes $j$ and $k$.[1]

## 1.1 Related Work

We provide a brief overview of approaches aiming to define graphical models for continuous random variables that go beyond the Gaussian case, and also of prior work on covariate dependent graphical models.

**Non-Gaussian Graphical models.** One line of work focuses on graphical models defined by monotone transformations of the Gaussian case, namely the non-paranormal (Liu et al., 2009; 2012). Their estimation entails first learning in a nonparametric fashion the respective transformations that "Gaussianize" the data, then subsequently leveraging approaches used for the Gaussian graphical model. Another broader line of work focuses on defining graphical models through specification of node conditional distributions. Yang et al. (2014) consider pairwise node conditional distributions from the exponential family, wherein the canonical parameter of the distribution is defined as a linear combination of higher order products of univariate functions of neighbors of each node $j$ in $G$. This construction leads to a family of potential functions that defines a proper graphical model under $G$ for the joint distribution $\mathbb{P}(\boldsymbol{x})$ (see details in Appendix D.2).

---

[1]Note that in the case of a Gaussian distributed error term $\varepsilon$, a *covariate* dependent graphical model is defined in terms of the joint distribution of $\boldsymbol{x}$ conditional on $\boldsymbol{z}$, i.e., $\mathbb{P}(\boldsymbol{x}|\boldsymbol{z})$, that factorizes over a graph $G(\boldsymbol{z})$, "parameterized" by the external covariate $\boldsymbol{z}$.

More recently, Baptista et al. (2024b) introduce a framework wherein the conditional dependence/independence between $x_j$ and $x_k$ is characterized by a score matrix given by $\Omega_{jk} := \mathbb{E}_{p(\boldsymbol{x})}[\partial_j \partial_k \log p(\boldsymbol{x})]^2$, where $p(\boldsymbol{x})$ is the density function of $\boldsymbol{x} \sim \mathbb{P}(\boldsymbol{x})$ and is further assumed to be continuously differentiable. The authors propose to estimate the density $p(\boldsymbol{x})$ via a lower triangular transport map, with the components of the map parameterized through linear expansions with Hermite functions as the bases. The same score matrix $\Omega$ is also leveraged in Zheng et al. (2023); however, instead of estimating the density $p(\boldsymbol{x})$, their approach parameterizes the data score $\nabla_{\boldsymbol{x}} \log p(\boldsymbol{x})$ using neural networks and estimates it based on an implicit score matching objective (Hyvärinen, 2005), with an additional sparsity-inducing term associated with $\Omega$.

**Covariate-dependent graphical models.** At a high level, covariate-dependent graphical models fall under the umbrella of "varying coefficient models", where model parameters are assumed to evolve with "dynamic features". Traditionally, the specific functional dependency on these dynamic features is handled via polynomial or smoothing splines, as well as kernel-local polynomial smoothing (see Fan & Zhang, 2008, and references therein). More recently, it has been rebranded by Al-Shedivat et al. (2020, CEN), where dynamic features are termed "contextual information". Specifically, the authors consider a hierarchical probabilistic framework in which the model parameters depend non-deterministically on the contextual information, and the dependency is parameterized through deep neural networks.

Within the specific context of undirected graphical model estimation where the focus is on conditional independence relationships, the existing literature can be broadly segmented into two categories: (1) the covariate is a categorical variable taking discrete values. Early work by Lafferty et al. (2001) considers a conditional random field (CRF) for discrete $\boldsymbol{x}$, and further assuming the underlying graph to be a chain graph. For high-dimensional settings, with a continuous $\boldsymbol{x}$, the problem is cast as efficient estimation of graphical models while taking into account their similarities across different categories (Guo et al., 2011; Cai et al., 2016). (2) For more general dependency on covariates that can be continuous-valued, one line of work focuses on partition-based estimation, namely, segmenting the samples based on the values of the covariates, then performing estimation within each partition (Liu et al., 2010). More recently, Gaussian graphical models, in which the graph structure is learned as a continuous function of the covariates, are being actively investigated. Specifically, Zhang & Li (2023); Zeng et al. (2024) assume a *linear* functional dependency on the covariate, namely, $\beta_{jk}(\boldsymbol{z}) = \boldsymbol{b}_{jk}^{\top}\boldsymbol{z}$, with $\boldsymbol{b}$ being sparse. The former adopts a regularization-based approach, while the latter a Bayesian one with sparsity inducing prior distributions. Within the Bayesian paradigm, notable contributions also include Ni et al. (2022); Niu et al. (2024) that introduce covariate-dependent prior distributions to enable context-dependent precision matrix estimation. Finally, we remark that graphical model estimation with non-linear dependency on the covariate has been investigated leveraging the idea of kernel smoothing (e.g., Zhou et al., 2010; Kolar et al., 2010). However, these approaches are limited to the case where the covariate is *univariate*, due to limitations induced by smoothing kernels. Table 1 presents a selective summary of the above-mentioned existing works, with an emphasis on those in which the covariate takes continuous values and corresponding theoretical results provided. We note that the score matrix-based formulation for estimating non-Gaussian graphical models could potentially be extended to incorporate the dependency on covariates, and such an extension is briefly discussed in Section 6.

As a concluding remark, we note that "contextualized" estimation has also been adopted in other settings, such as the estimation of directed acyclic graphs (DAGs); see, e.g., Zhou et al. (2022); Thompson et al. (2024) and references therein.

## 1.2 Our Contributions

This work differs from existing ones in that it investigates a deep neural network-based approach for estimating covariate-dependent graphical models, and establishes the corresponding statistical guarantees under the PAC learning framework. Unlike much of the above-mentioned statistical literature that assumes a linear relationship with the covariate, the current work leverages parameterization via neural networks to accommodate more flexible dependency structures. On the theoretical front, finite-sample error bounds are established under certain regularity conditions, enabling the quantification of the relationship between the size of the neural network and the estimation error. Note that although Al-Shedivat et al. (2020) outline

Table 1: Summary of representative works in extant literature on covariate-dependent graphical models.

| | | |
|---|---|---|
| Zhou et al. (2010) | setup | univariate, Gaussian, MLE-based kernel estimator |
| | theor.res. | finite-sample Frobenius and large-deviation bound |
| Kolar et al. (2010) | setup | univariate, nodewise regression-based kernel smoothing |
| | theor.res. | finite-sample support recovery consistency under sub-Gaussanity and minimum edge strength assumption |
| Liu et al. (2010) | setup | multivariate, Gaussian, tree-based partition & MLE within the region |
| | theor.res. | guarantee on the excess risk of the estimator |
| Zhang & Li (2023) | setup | multivariate, Gaussian, nodewise regression w/ linear dependency on covariate |
| | theor.res. | finite-sample $\ell_2$ consistency and support recovery consistency |
| Ni et al. (2022) | setup | multivariate, Gaussian, likelihood-based w/ linear dependency on covariate |
| (Bayesian) | theor.res. | no guarantee on the posterior inference of the estimates provided |
| Niu et al. (2024) | setup | multivariate, covariate-dependent priors & Gaussian likelihood-based estimation (within partition) |
| (Bayesian) | theor.res. | convergence of the posterior distribution in an $\alpha$-divergence metric |
| **This Work** | setup | multivariate, nodewise regression with flexible dependency on covariate, parameterized by DNN |
| | theor.res. | finite-sample $\ell_2$ consistency and edge-wise recovery guarantee with thresholding |

Note: for `setup`, univariate (or multivariate) corresponds to the dimension of the covariate $\boldsymbol{z}$ in question, whereas "Gaussian" corresponds to the distributional assumption for $\boldsymbol{x}$ if explicitly assumed in the paper.

a generic template for handling contextual information via deep neural networks, the theoretical results therein primarily focus on (i) quantifying the contribution of the context to predictive performance, under the assumption that the expected predictive accuracy is bounded below by $1 - \varepsilon$ (Proposition 4), and (ii) analyzing the error bound of the estimated linear coefficient in binary classifications tasks (Theorem 6). Both results are established under the assumption that the underlying true dependency on the covariate is *linear*.

To this end, the contribution of this work can be summarized as follows.

- We consider a nonlinear covariate-dependent graphical model, which extends existing ones and allows for flexible functional dependency on the covariate. Empirically, we demonstrate that such an estimation framework effectively learns the dependency of the graph structure on the covariate $\boldsymbol{z}$; meanwhile, it accommodates complex conditional dependence structures amongst the $\boldsymbol{x}$ and exhibits good performance even when the underlying data deviates from Gaussianity.

- On the theoretical front, we establish guarantees on the recovered edges based on a deep neural network estimator. Concretely, a mis-specified setting[2] is considered wherein both a generalization error and an approximation error are present, and their bounds depend on the size of the underlying problem (namely $p$ and $q$), the size of the neural network and the sample size.

  In particular, this work illustrates how existing results—specifically, generalization error bounds based on local Rademacher complexity analysis (Bartlett et al., 2005) and approximation error bounds of an MLP (Kohler & Langer, 2021)—established under generic ERM settings, can be synthesized to obtain error bounds for a specific estimation problem. Such a roadmap is of interest for similar inference tasks.

Note that the roadmap for the technical analysis presented in this paper applies to a broad class of strongly convex, Lipschitz, and uniformly bounded loss functions, by connecting the infinity norm of the function approximation in Kohler & Langer (2021) to the excess error bound in Bartlett et al. (2005) through a strong convexity-based argument. Hence, it goes beyond the result established for the mean squared error loss function studied in Kohler & Langer (2021).

## 2  DNN-based Covariate-Dependent Graphical Model Estimation

We elaborate on the model under consideration in (1) and provide its formal definition next. We consider a system of $p$ variables $\boldsymbol{x} := (x_1, x_2, \cdots, x_p)'$ taking values in $\mathcal{X} \subseteq \mathbb{R}^p$, whose conditional dependence structure depends on some $q$-dimensional "external" covariate $\boldsymbol{z} \in \mathcal{Z} \subseteq \mathbb{R}^q$. Let $[p] := \{1, \cdots, p\}$ denote the index

---

[2]Here mis-specified is in the ERM context, where the target hypothesis does not necessarily live in the hypothesis class in which empirical minimization is conducted.

set of the variables. To model such conditional dependence structures, we consider a nodewise regression formulation as the *working model*, given in the form of

$$x_j = \langle \boldsymbol{\beta}_j(\boldsymbol{z}), \boldsymbol{x}_{-j} \rangle + \varepsilon_j := \sum_{k \neq j}^{p} \beta_{jk}(\boldsymbol{z}) x_k + \varepsilon_j; \quad j \in [p]. \tag{2}$$

$\mathbb{E}(\varepsilon_j) = 0$ and has finite moments; $\beta_{jk}(\cdot) : \mathcal{Z} \mapsto \mathbb{R}$ is a function that takes $\boldsymbol{z}$ as the input and outputs the regression coefficients for $x_k$'s with $x_j$ being the response. For notation simplicity, we let $\boldsymbol{\beta}_j(\boldsymbol{z}) : \mathcal{Z} \mapsto \mathbb{R}^{p-1}$, whose output coordinates correspond to $\beta_{jk}(\boldsymbol{z}), k = 1, \cdots, p, k \neq j$, and $\boldsymbol{x}_{-j} := (x_1, \cdots, x_{j-1}, x_{j+1}, \cdots, x_p)'$. We note the analogy to the node-wise regression formulation for the Gaussian graphical model (Meinshausen & Bühlmann, 2006) (see also Appendix D.1 for a brief discussion), where in the absence of the external covariate $\boldsymbol{z}$, the regression coefficient $\beta_{jk}$ "degenerates" to a scalar. With the dependency on $\boldsymbol{z}$, the graphical model is solely captured in the $\boldsymbol{\beta}_j(\boldsymbol{z})$'s, and one can retrieve the estimated graph at the *sample* level, based on the corresponding $\boldsymbol{z}$ value for the sample of interest. In particular, when $\boldsymbol{x}$ is jointly Gaussian, the node-conditional distribution is given by $\mathbb{P}(x_j | \boldsymbol{x}_{-j}, \boldsymbol{z}) \sim \mathcal{N}(\sum_{k \neq j}^{p} \beta_{jk}(\boldsymbol{z}) x_k, \sigma_j^2(\boldsymbol{z}))$.

We consider parameterizing each $\beta_{jk}(\cdot)$ by some neural network, e.g., a multi-layer perceptron (MLP). Let $\boldsymbol{\theta}$ denote the parameters of the neural networks in question collectively. Given independently identically distributed (i.i.d.) samples $\{(\boldsymbol{x}^i, \boldsymbol{z}^i)\}_{i=1}^{n}$, $\boldsymbol{\theta}$ can be obtained by minimizing the empirical loss; e.g., with a mean squared error loss, it is given by

$$\arg \min_{\boldsymbol{\theta}} \frac{1}{n} \sum_{i=1}^{n} \|\boldsymbol{x}^i - \widehat{\boldsymbol{x}}^i\|_2^2, \text{ where } \widehat{x}_j^i = \langle \boldsymbol{\beta}_j(\boldsymbol{z}^i; \boldsymbol{\theta}), \boldsymbol{x}_{-j}^i \rangle; \tag{3}$$

the dependency on the neural network parameter $\boldsymbol{\theta}$ is made explicit. The training pipeline is outlined in Exhibit 1.

---

**Exhibit 1:** DNN-based Covariate-dependent Graphical Model (**DNN-CGM**) Learning Pipeline

**Input:** Input i.i.d. samples $\{(\boldsymbol{x}^i, \boldsymbol{z}^i)\}_{i=1}^{n}$

1. [`forward pass`] for fixed $\boldsymbol{\theta}$, calculate $\widehat{x}_j^i$ for all $j = 1, \cdots, p$ and $i = 1, \cdots, n$ according to (3);

2. [`loss`] calculate the loss based on all samples, by evaluating the distance between $\boldsymbol{x}^i$ and $\widehat{\boldsymbol{x}}^i$;

3. [`backward pass`] update $\boldsymbol{\theta}$ using the gradients calculated based on the loss (back-propagation);

**Output:** Estimated $\widehat{\boldsymbol{\theta}}$ and $\{\boldsymbol{\beta}_j(\boldsymbol{z}^i, \widehat{\boldsymbol{\theta}})\}_{j=1}^{p}, \forall i$

---

*Remark* 1 (On the implementation of $\beta_{jk}(\cdot)$ networks). Instead of creating separate neural networks for each $\beta_{jk}(\cdot)$—which requires $p(p-1)$ networks, with each taking $\boldsymbol{z}$ as the input and generating a scalar output—one can alternatively create neural network(s) with appropriately sized hidden/output layers so that "backbones" are shared across and fewer networks are needed. This leads to fewer total number of parameters needed and empirically an easier optimization problem, which often yields superior performance. As a concrete example where a single neural network is used, it takes $\boldsymbol{z}$ as the input and outputs a $p(p-1)$ dimensional vector, with each coordinate corresponding to the value of $\beta_{jk}(\boldsymbol{z}), j, k \in [p]; k \neq j$; such a design corresponds to the case where $\beta_{jk}(\boldsymbol{z})$ is operationalized as $\beta_{jk}(\boldsymbol{z}) = v_{jk}(u(\boldsymbol{z}))$. Specifically, let $h$ be the dimension of the last hidden layer; $u(\cdot) : \mathbb{R}^q \mapsto \mathbb{R}^h$ is shared across all $\beta_{jk}$'s and corresponds to stacked neural network layers up until the last hidden layer; $v_{jk}(\cdot) : \mathbb{R}^h \mapsto \mathbb{R}$ is a linear layer that maps neurons from the last hidden layer to the output unit indexed by $jk$, and it differs across the $jk$'s. In this case, $\boldsymbol{\theta} = (\boldsymbol{\theta}_u, \{\boldsymbol{\theta}_{v,jk}, j, k \in [p]; k \neq j\})$.

*Remark* 2. The formulation under consideration is not restricted to $\boldsymbol{z}$ being continuous. In the case where any coordinate(s) of $\boldsymbol{z}$ is categorical, one can first process the categories via an embedding layer, then concatenate them with the numerical ones and proceed.

Some considerations on the formulation are discussed next. First note that for a node-conditional formulation where the conditional distribution of each node is some function of its neighborhood set (Meinshausen & Bühlmann, 2006), the model in its most general form can be written as $x_j = f_j(\boldsymbol{x}_{-j}, \varepsilon_j), j \in [p]$. Such a form can be too general, and potentially incur difficulty in actually identifying the neighborhood set based

on data[3]. In the same spirit as in Bühlmann et al. (2014), by considering an additive form, the model is restricted to $x_j = \sum_{k \neq j} f_{jk}(x_k) + \varepsilon_j$; in particular, $f_{jk}(x_k) = 0$ for $k \notin \text{ne}(j)$; $\text{ne}(j) \subseteq [p] \setminus \{j\}$ is the neighborhood of node $j$. With the presence of the covariate $\boldsymbol{z}$ that impacts the neighborhood structure, a natural extension is to augment the input space of $f_{jk}$'s, namely,

$$x_j = \sum_{k \neq j} f_{jk}(\boldsymbol{z}, x_k) + \varepsilon_j; \quad j \in [p].$$

In this paper, we impose additional structure on the specification of $f_{jk}$ and let $f_{jk}(\boldsymbol{z}, x_k) \equiv \beta_{jk}(\boldsymbol{z}) x_k$. The implication is two-fold:

1. $\boldsymbol{z}$ and $x_k$ are assumed separable. In particular, we consider a multiplicative form, namely, $f_{jk}(\boldsymbol{z}, x_k) \equiv \beta_{jk}(z) \gamma_{jk}(x_k)$, that behaves like a "locally linear" model on $\gamma_{jk}(x_k)$, and $\beta_{jk}(\boldsymbol{z})$ can be viewed as a *generalized regression coefficient*. The edge weight can be summarized by $\beta_{jk}(\boldsymbol{z})$, which can be interpreted as the strength of relevance; in addition, $\beta_{jk}(\boldsymbol{z}) = 0 \Rightarrow f_{jk}(\boldsymbol{z}, x_k) = 0$. This preserves the interpretability of the model in that the neighborhood set can be solely inferred based on $\{\beta_{jk}(\boldsymbol{z})\}_{j=1,\cdots,p;k \neq j}$, a primary quantity of interest that can be directly estimated and a probabilistic guarantee can be established accordingly. Note that for $f_{jk}(\boldsymbol{z}, x_k)$ that allows for general interactions between $\boldsymbol{z}$ and $x_k$, extracting the underlying *weighted* graph edge that characterizes the *strength* of the dependency requires post-hoc operations. For example, this can be achieved by considering quantities of the form $\nabla_{x_k} f_{jk}(\boldsymbol{z}, x_k)$.

   A similar separable structure is employed in Alvarez Melis & Jaakkola (2018); Marcinkevičs & Vogt (2021), where, despite a slightly different model setup that involves *only* $\boldsymbol{x}$, the function is decomposed as $\theta(\boldsymbol{x}) h(\boldsymbol{x})$, where $h(\boldsymbol{x})$ serves as a representation of $\boldsymbol{x}$ and $\theta(\boldsymbol{x})$ its corresponding relevance; those models are interpretable and "self-explaining".

2. The functions $\gamma_{jk}(x_k)$'s are further simplified to $\gamma_{jk}(x_k) \equiv x_k$. From a regression analysis perspective, $x_k$ and its transformations can be interpreted as features. We use "raw" features directly, partly due to the fact for the graphical model setting under consideration, they correspond to node values and thus are inherently meaningful. This contrasts with, for example, image pixels that lack intrinsic intepretability and thus benefit from transformations into higher level features (Alvarez Melis & Jaakkola, 2018). One can potentially consider a more complex $\gamma_{jk}(\cdot)$, e.g., parameterize it by some neural network, or through basis expansion (e.g., Qiao et al., 2019), namely, $\gamma_{jk}(x) := \sum_{\ell=1}^{L} a_{jk,\ell} \phi_\ell(x)$ with the bases $\phi_\ell(\cdot)$ fixed apriori and $a_{jk,\ell}$ learnable parameters. Empirically, we observe that the current specification is essentially as powerful as its enriched counterparts (modulo tuning), in terms of identifying the neighborhood of each node.

*Remark* 3. Up to this point, we do not impose any sparsity assumptions on the underlying graph and the estimation procedure does not involve any regularization terms. In practice, to obtain a sparsified graph for interpretability purposes, we consider a thresholding procedure, where the small entries can be effectively treated as zeros. The thresholding level is selected by examining the edge magnitude histogram, and chosen in the region where a "gap" is present[4]. Finally, given the undirected nature of the graph, we adopt the "AND" principle, namely, we consider an edge as present when both $\widehat{\beta}_{jk}$ and $\widehat{\beta}_{kj}$ are nonzero after hard-thresholding.

## 3 Theoretical Results

This section provides theoretical guarantees for the model posited in (2), assuming that it corresponds to the ground-truth data generating process (DGP). Specifically, let $\boldsymbol{\beta}^*(\cdot)$ denote the target model (i.e., the true DGP one); given the availability of i.i.d. data $\{(\boldsymbol{x}^i, \boldsymbol{z}^i)\}_{i=1}^n$ generated according to (2), we are interested in the

---

[3]With a general $f_j$ where interactions amongst the $x_k$'s are permitted, for $x_j \perp\!\!\!\perp x_k \,|\, \boldsymbol{x}_{-k}$ to hold, $x_k$ needs to be absent from *all* terms, which is difficult to operationalize either through constraints or in a post-hoc fashion.

[4]See also the setting considered in Corollary 2. Note that in the case where the underlying true graph has edges that can be separated into weak and strong ones (with exact sparsity being a special case), one would expect a gap in the histogram, aside from estimation/approximation errors. Empirically, one can first "normalize" the graph so that the maximum entry does not exceed 1 in magnitude; the thresholding level is then scaled accordingly. This normalization step can potentially enhance the interpretability of the the thresholding level, without fundamentally altering the sparsification outcome.

error bound of $\widehat{\boldsymbol{\beta}}(\cdot)$ under the Empirical Risk Minimization (ERM) framework, with $\beta_{jk}(\cdot)$'s parameterized by deep neural networks. We establish finite-sample bounds for the *generalization error* using Local Rademacher complexity tools (Bartlett et al., 2005), and the *approximation error bound* in terms of neural network hyper-parameters (namely, depth and number of neurons) leveraging the techniques in Kohler & Langer (2021), under certain regularity conditions. The two combined provides insight into consistency type of results for $\widehat{\boldsymbol{\beta}}(\cdot)$, relative to the ground truth $\boldsymbol{\beta}^*(\cdot)$. In the ensuing technical developments, we further assume that the random vectors $\boldsymbol{x}, \boldsymbol{z}$ are bounded, namely $\boldsymbol{x} \in [-C, C]^p$ and $\boldsymbol{z} \in [-C, C]^q$, respectively. Without loss of generality, we set $C = 1$, which only affects the scale in the analysis.

## 3.1  Preliminaries and A Road Map

**Definitions.**   Let $(\mathcal{Y}, \mathbb{P})$ denote a probability space and $\mathbb{E}_{\boldsymbol{y}}[\cdot]$ denote the expectation with respect to $\mathbb{P}$ over random variable $\boldsymbol{y} \in \mathcal{Y}$. Let $\mathcal{H}$ denote a hypothesis class; i.e., a class of measurable functions $h : \mathcal{Y} \mapsto \mathbb{R}$. The infinity norm between two functions $h, \tilde{h}$ is defined as $\|h - \tilde{h}\|_\infty := \sup_{\boldsymbol{y} \in \mathcal{Y}} |h(\boldsymbol{y}) - \tilde{h}(\boldsymbol{y})|$, while for vector functions $\boldsymbol{h} = (h_1, \cdots, h_p)$, the corresponding vectorized infinity norm as $\|\boldsymbol{h} - \tilde{\boldsymbol{h}}\|_\infty := \max_{j \in \{1, \cdots, p\}} \|h_j - \tilde{h}_j\|_\infty$. For two functions, $h, g$, the notation $h \gtrsim g$ means that $h \geq cg$ for some universal constant $c$, and $\lesssim$ is analogously defined. Further, $g \simeq h$, if $\underline{c}g \leq h \leq \bar{c}g$ for some universal constants $\bar{c}$ and $\underline{c}$. Finally, for a real-valued vector $\boldsymbol{y}$, its weighted norm with respect to a matrix $A$ is given by $\|\boldsymbol{y}\|_A^2 := \langle \boldsymbol{y}, A\boldsymbol{y} \rangle$.

For the problem at hand, let $\boldsymbol{y} := (\boldsymbol{x}, \boldsymbol{z})$; $S_n := \{(\boldsymbol{x}^i, \boldsymbol{z}^i)\}_{i=1}^n \subset (\mathcal{X} \times \mathcal{Z})^n$ denote a random sample of $n$ iid data points. We consider a loss function $\ell(a; b) : \mathbb{R} \times \mathbb{R} \mapsto \mathbb{R}$ and $\mathcal{H}$ the hypothesis class of risk functions of the form $h(\boldsymbol{y}) := (\ell \circ \boldsymbol{\beta})(\boldsymbol{y}) := \frac{1}{p} \sum_{j=1}^p \ell\big(\sum_{k \neq j} \beta_{jk}(\boldsymbol{z}) x_k, x_j\big)$, i.e., the risk is a composite function of the loss and regression coefficients. Then, the population and empirical versions of the risk are respectively defined as follows, with $[n] := \{1, \cdots, n\}$:

$$\mathcal{R}(\boldsymbol{\beta}) := \mathbb{E}_{\boldsymbol{x}, \boldsymbol{z}} \left[ \frac{1}{p} \sum_{j=1}^p \ell\left( \langle \boldsymbol{\beta}_j(\boldsymbol{z}), \boldsymbol{x}_{-j} \rangle, x_j \right) \right]; \tag{4}$$

$$\mathcal{R}_n(\boldsymbol{\beta}) := \frac{1}{n} \sum_{i \in [n]} \left[ \frac{1}{p} \sum_{j=1}^p \ell\left( \langle \boldsymbol{\beta}_j(\boldsymbol{z}^i), \boldsymbol{x}_{-j}^i \rangle, x_j^i \right) \right].$$

To perform ERM, we assume that the regression coefficient functions $\beta_{jk}(\cdot) : \mathcal{Z} \mapsto \mathbb{R}$ belong to some hypothesis class $\mathcal{F}$[5], and denote $\mathcal{F}_{p \times (p-1)}$ as the joint of $p \times (p-1)$ hypotheses, with each individual hypothesis $\beta_{jk}(\cdot) \in \mathcal{F}, j, k \in [p], k \neq j$. We further define a collection of optimal hypotheses $\beta_{jk}^{\text{opt}}$'s as

$$\beta_{jk}^{\text{opt}} := \arg\min_{\beta_{jk} \in \mathcal{F}} \|\beta_{jk} - \beta_{jk}^*\|_\infty, \quad \forall j, k \in [p]; k \neq j,$$

and the approximation error can be defined as $\mathcal{E}_{\text{approx}}(\mathcal{F}) := \max_{j \in \{1, \cdots, p\}} \|\boldsymbol{\beta}_j^{\text{opt}} - \boldsymbol{\beta}_j^*\|_\infty$. Finally, the ERM estimator of the regression coefficient functions is given by

$$\widehat{\boldsymbol{\beta}} := \arg\min_{\boldsymbol{\beta} \in \mathcal{F}^{p \times (p-1)}} \mathcal{R}_n(\boldsymbol{\beta}). \tag{5}$$

**Assumptions.**   Before presenting the results, we posit the assumptions used in the theoretical analysis. Together with the boundedness assumption on $(\boldsymbol{x}, \boldsymbol{z})$ mentioned above, these assumptions lead to a $O(1/n)$ fast convergence rates for the risk excess error, and hence a smaller sample size requirement. Further, the analysis enables us to balance the trade-off between approximation error and generalization error based on the size of the neural network, the sample size and the dimension of $(\boldsymbol{x}, \boldsymbol{z})$.

**Assumption 1** (On the loss function). *The loss function $\ell(a; b) : \mathbb{R} \times \mathbb{R} \mapsto [0, M]$; w.o.l.g. we set $M = 1$[6]. We assume $\ell(a, b)$ is:*

- *$L$-Lipschitz in $a$: $|\ell(a_1; b) - \ell(a_2; b)| \leq L|a_1 - a_2|$, where $L > 0$ denotes the Lipschitz constant.*

- *$\alpha$-strongly convex in $a$: $\ell(a_1; b) - \ell(a_2; b) \geq \ell'(a; b)_{|a=a_2}(a_1 - a_2) + \frac{\alpha}{2}(a_1 - a_2)^2$ where $\alpha > 0$ and $\ell'(\cdot)$ denotes the derivative of the function $\ell$.*

---

[5]$\mathcal{F}$ is typically selected by practitioners, such as neural networks with fixed number of layers and neurons.
[6]For any loss function $\ell(a; b) : \mathbb{R} \times \mathbb{R} \mapsto [0, M]$, it suffices to consider its rescaled version $\frac{1}{M} \cdot \ell(a; b)$ to satisfy such an assumption.

*Remark* 4. Assumption 1 is widely used in the statistical learning literature, especially when aiming to establish fast rates for the generalization error bound (see, e.g., Klochkov & Zhivotovskiy, 2021).

**Assumption 2** (On the optimality of $\beta_{jk}^*$). *The optimal regression coefficient function $\beta_j^*, j \in [p]$ satisfies the following condition:*

$$\mathbb{E}_{\boldsymbol{x},\boldsymbol{z}}\left[\sum_{j=1}^p \ell'(\langle \boldsymbol{\beta}_j(\boldsymbol{z}), \boldsymbol{x}_{-j}\rangle, x_j)_{|\boldsymbol{\beta}_j = \boldsymbol{\beta}_j^*}\right] = 0.$$

*Remark* 5. Note that the combination of Assumptions 1 and 2 implies that $\beta_{jk}^*, j, k \in [p], k \neq j$ minimizes the population risk in (4) due to the strong convexity of the loss function $\ell$. We also demonstrate that the mean square error loss function satisfies Assumptions 1 and 2 in Appendix A.2.

**Assumption 3** (Bounded pseudo-dimension of hypothesis class $\mathcal{F}$). *The hypothesis class $\mathcal{F}$ of the regression coefficient functions $\boldsymbol{\beta}_j, j \in [p]$ has finite pseudo-dimension (Pollard, 1990); i.e., $d_P(\mathcal{F}) < \infty$ (see formal definition of pseudo-dimension in Definition 3 in Appendix A.1).*

This assumption is widely adopted in the statistical learning literature and encompasses a broad range of functional classes, including linear and polynomial functions (Anthony & Bartlett, 1999), as well as various families of neural networks (Bartlett & Maass, 2003; Bartlett et al., 2019; Khavari & Rabusseau, 2021).

The primary result established for the model postulated in (1) is the statistical consistency for the estimate $\widehat{\boldsymbol{\beta}}$ (Corollary 1). This is achieved through two key steps: (i) derive a bound on the excess error, defined as $\mathcal{R}(\widehat{\boldsymbol{\beta}}) - \mathcal{R}(\boldsymbol{\beta}^*)$, as a function of $\widehat{\boldsymbol{\beta}}$; see Theorem 1; and (ii) leveraging the strong convexity of the loss function (mean squared error loss) and the result in Theorem 1, derive the error bound for $\widehat{\boldsymbol{\beta}}$ under some projection norm. The bound in Theorem 1 has two components that respectively capture the generalization error and the approximation error, with the latter arising from the mis-specified setting assumption that allows the true $\boldsymbol{\beta}^*$ to live outside of the working hypothesis class $\mathcal{F}$. To derive a bound for the generalization error, we first show that the hypothesis class—that encompasses composite functions quantifying the "delta in risk" between $\boldsymbol{\beta}$ and the true $\boldsymbol{\beta}^*$—satisfies Bernstein's Condition (Lemma 1). Next, we select an appropriate subroot function $\tau(\cdot)$ tailored to the problem, enabling us to apply the proof strategy and results from Bartlett et al. (2005) (specifically, Theorem 3.3, Corollary 2.2, and Corollary 3.7). A critical component of the proof involves bounding the covering number of the function class $\beta_{jk}(\boldsymbol{z})$ under consideration.

With the working hypothesis class encompassing a class of MLPs, Theorem 2 characterizes the approximation error for such architectures, employing tools from Kohler & Langer (2021). Note that the pseudo-dimension—which appears in the generalization error bound—can also be associated with the size of the neural network. Consequently, the two error components for the MLP architecture under consideration can be "balanced", in that one can minimize the excess error bound by choosing the quantity that links the two appropriately.

## 3.2   Main Results

The first result leverages Local Rademacher Complexity and empirical processes tools to bound the generalization error:

$$\underbrace{\mathcal{R}(\widehat{\boldsymbol{\beta}}) - \mathcal{R}(\boldsymbol{\beta}^*)}_{\text{excess error}} \lesssim \underbrace{\mathcal{R}_n(\widehat{\boldsymbol{\beta}}) - \mathcal{R}_n(\boldsymbol{\beta}^*)}_{\text{empirical excess error}} + \underbrace{O\left(\frac{d_P(\mathcal{F})}{n}\right)}_{\text{generalization error}},$$

where the $O(\cdot)$ notation ignores all problem dependent constants and only depends on $n$ and $d_P(\mathcal{F})$. The empirical excess error can be further bounded by an approximation error term controlled by the capacity of the hypothesis class (see equation (15) in the proof of Theorem 1 in Appendix A.3).

**Theorem 1.** Under Assumptions 1-3, the following bound for the excess error as a function of the empirical risk minimizer $\widehat{\boldsymbol{\beta}}$ (see (5)) based on $S_n$ samples holds with probability at least $1 - \delta$, for any $\delta > 0$:

$$\mathcal{R}(\widehat{\boldsymbol{\beta}}) - \mathcal{R}(\boldsymbol{\beta}^*) \lesssim \underbrace{L \cdot \mathcal{E}_{\text{approx}}(\mathcal{F})}_{\text{approximation error}} + \underbrace{\frac{L^2 d_P(\mathcal{F})}{\alpha n} p^2 \log(nLp) \log\left(\frac{1}{\delta}\right)}_{\text{generalization error}}. \tag{6}$$

*Remark* 6. The $p^2$ term in (6) shows up due to estimating $p \times (p-1)$ functions using neural networks, which naturally increases the entropy number of $\mathcal{F}$ with respect to $S_n$ in Dudley's entropy integral (Dudley, 2016) by a factor of $p^2$. This indicates that the size of the graphical model can grow at a $p = o(\sqrt{n})$ rate for the excess error to vanish. A larger $p$ can be accommodated, either by incorporating sparsity assumptions (Meinshausen & Bühlmann, 2006; Wainwright, 2009) or through refined analysis as in multi-task learning (Lounici et al., 2009), which however is outside the scope of the current paper.

The following corollary establishes the consistency of the empirical risk minimizer, leveraging the strong convexity of the loss function.

**Corollary 1** (Consistency of $\widehat{\boldsymbol{\beta}}$)**.** Let $A_j(\boldsymbol{z}) := \mathbb{E}_{\boldsymbol{x}}\big[\boldsymbol{x}_{-j}\boldsymbol{x}_{-j}^\top \big| \boldsymbol{z}\big]$. Under the Assumptions of Theorem 1, the following inequality holds with probability at least $1 - \delta$ for any $\delta > 0$:

$$\mathbb{E}_{\boldsymbol{z}}\left[\frac{1}{p}\sum_{j=1}^{p}\|\widehat{\boldsymbol{\beta}}_j(\boldsymbol{z}) - \boldsymbol{\beta}_j^*(\boldsymbol{z})\|_{A_j(\boldsymbol{z})}^2\right] \leq \frac{L^2 d_P(\mathcal{F})}{\alpha^2 n}p^2 \log(nLp)\log\left(\frac{1}{\delta}\right) + \frac{L \cdot \mathcal{E}_{\mathrm{approx}}(\mathcal{F})}{\alpha}. \tag{7}$$

Corollary 1 quantifies the quality of the recovery of the regression coefficient functions $\boldsymbol{\beta}_j$ that capture the strength of the conditional dependence relationships between the variables.

Note that to establish Theorem 1, the assumption of *realizability*—i.e., $\beta_{jk}^*(\cdot) \in \mathcal{F}$—is not imposed. Consequently, the empirical risk of the ERM estimator is not guaranteed to be smaller than that of $\boldsymbol{\beta}^*$, i.e., $\mathcal{R}_n(\boldsymbol{\beta}^*) \leq \mathcal{R}_n(\widehat{\boldsymbol{\beta}})$ is not guaranteed. In particular, this can happen when the hypothesis class does not have adequate capacity, and hence the approximation error term is nonzero and appears in (6). However, Theorem 1 has not characterized the approximation error in terms of the sample size and dimension $p$ of the graphical model. This is established in Theorem 2, which also aims to balance the approximation error and generalization error terms.

To that end, in addition to Assumptions 1-3 used to derive the excessive risk bound, we need the following additional assumption to apply tools from Kohler & Langer (2021) to quantify the approximation error based on the number of layers and the total number of neurons used by the feed-forward neural networks to approximate the regression coefficients $\beta_{jk}$.

**Assumption 4** $((m, C)$-smoothness of $\beta_{jk}^*$)**.** *We assume $\beta_{jk}^*$ is $(m, C)$-smooth for all $j, k \in [p], k \neq j$ (see the formal Definition 4 in Appendix A.1).*

**Theorem 2.** Let $\mathcal{F}$ be a family of fully connected neural networks with ReLU activation functions, having number of layers $H \simeq \xi^{-q/2m}$ and number of neurons $r \simeq (2e)^q \binom{m+q}{q}q^2$. Under the Assumptions of Theorem 1, together with Assumption 4 with $m \lesssim q$, by setting

$$\xi = \left(\frac{Lm^4 d^6 p^2 \log^2(nLp)\log(p)\log(1/\delta)\log(1/\alpha)}{\alpha n}\right)^{\frac{m}{m+q}},$$

the following holds with probability at least $1 - \delta$:

$$\mathcal{R}(\widehat{\boldsymbol{\beta}}) - \mathcal{R}(\boldsymbol{\beta}^*) \lesssim L\xi, \quad \text{and} \quad \mathbb{E}_{\boldsymbol{z}}\left[\frac{1}{p}\sum_{j=1}^{p}\|\widehat{\boldsymbol{\beta}}_j(\boldsymbol{z}) - \boldsymbol{\beta}_j^*(\boldsymbol{z})\|_{A_j(\boldsymbol{z})}^2\right] \lesssim \frac{L\xi}{\alpha}. \tag{8}$$

*Remark* 7. The rates of the bounds in (8) are of the order $n^{-m/(m+q)}$, whereas the rate in Kohler & Langer (2021) based on a quadratic loss is $n^{-2m/(2m+q)}$. This discrepancy is due to the fact that when analyzing general $L$-Lipschitz functions, the excess error bound depends in a linear fashion on the approximation error of $\boldsymbol{\beta}(\boldsymbol{z})$. For the mean squared error loss function, it depends in a quadratic fashion and hence the same rate as in Kohler & Langer (2021) can be obtained, as shown in Appendix A.6.

**Extensions to edge-wise recovery.** Recall that in Corollary 1, an $\ell_2$-type consistency result is established for $\widehat{\boldsymbol{\beta}}$ based on a projection norm. If we consider a setting where edges can be segmented into strong and weak ones, then the result in Corollary 1 can be translated to an "edge-wise" type guarantee. Formally, we define the set of strong edges as $E^*(\boldsymbol{z}) := \{(j, k) \mid |\beta_{jk}^*(\boldsymbol{z})| \geq \underline{\beta}\}$ and of weak ones as $E^{*c}(\boldsymbol{z}) := \{(j, k) \mid |\beta_{jk}^*(\boldsymbol{z})| \leq \bar{\beta}\}$, where $\underline{\beta}$ and $\bar{\beta}$ correspond to their respective minimum/maximum magnitude. In addition, we assume that

there exists a uniform lower bound $\phi > 0$ on the margin between the maximum and minimum magnitudes; namely, $\underline{\beta} - \bar{\beta} \geq \phi > 0$. The edge-wise result in Corollary 2 focuses on establishing a guarantee that $\mathbb{1}\{|\widehat{\beta}_{jk}(\boldsymbol{z})| \geq \tau\}$ matches $\mathbb{1}\{|\widehat{\beta}^*_{jk}(\boldsymbol{z})| \geq \underline{\beta}\}$ in expectation, for some threshold $\tau := \eta\bar{\beta} + (1-\eta)\underline{\beta}$, $\eta > 0$; i.e., $\tau$ is a convex combination of $\bar{\beta}$ and $\underline{\beta}$.

**Assumption 5.** $A_j(\boldsymbol{z}) := \mathbb{E}\big[\boldsymbol{x}_{-j}\boldsymbol{x}_{-j}^\top|\boldsymbol{z}\big] \succ \gamma\boldsymbol{I}$ *holds uniformly for all* $\boldsymbol{z} \in \mathcal{Z}$ *with* $\gamma > 0$.

This assumption ensures that the minimum eigenvalue of $A_j(\boldsymbol{z})$ is bounded away from zero, uniformly for all $\boldsymbol{z}$. This assumption allows to us to first establish an analogous result as in (7), with the distance between $\widehat{\boldsymbol{\beta}}_j(\boldsymbol{z}) - \boldsymbol{\beta}^*_j(\boldsymbol{z})$ measured in the Euclidean norm $\|\cdot\|$ instead of the projection norm $A_j(\boldsymbol{z})$; subsequently, the Euclidean norm result is further translated to that on individual edges.

**Corollary 2.** Let $\tau := \eta\bar{\beta} + (1-\eta)\underline{\beta}$ with $\eta > 0$. Under the Assumptions of Corollary 1 and Assumption 5, the following inequality holds with probability at least $1 - \delta$:

$$\mathbb{E}_{\boldsymbol{z}}\Bigg[\sum_{j=1}^p \sum_{k \neq j} \mathbb{1}\big\{|\widehat{\beta}_{jk}(\boldsymbol{z})| \geq \tau\big\} \neq \mathbb{1}\big\{|\beta^*_{jk}(\boldsymbol{z})| \geq \underline{\beta}\big\}\Bigg] \lesssim \frac{L^2 d_P(\mathcal{F})}{\alpha^2 \min\{\eta^2, (1-\eta)^2\}\gamma(\underline{\beta} - \bar{\beta})^2 n} p^2 \log(nLp) \log\left(\frac{1}{\delta}\right)$$
$$+ \frac{L \cdot \mathcal{E}_{\text{approx}}(\mathcal{F})}{\alpha\gamma \min\{\eta^2, (1-\eta)^2\}(\underline{\beta} - \bar{\beta})^2}. \tag{9}$$

*Remark* 8. Some comments on the result in Corollary 2 are provided next.

- Similarly to the result in Corollary 1, the bound in (9) consists of two terms that correspond, respectively, to the generalization error and the approximation error bounds. With a constant margin $\underline{\beta} - \bar{\beta}$, the generalization error bound still vanishes, provided that the network size $p$ and the sample size $n$ grow at certain rates. However, the approximation error bound does not vanish; in particular, when the margin is small, the corresponding term can be large, resulting in a fairly loose bound. This aligns with intuition that when the strong and weak edges are hard to "distinguish", the edge-wise recovery via thresholding becomes difficult.

- The established bound is in expectation and can be interpreted as follows: on average, the strong edges can be identified via thresholding, provided that the threshold is selected within a certain range. This differs from the graph recovery results established for *sparse* high-dimensional Gaussian graphical models based on an $\ell_1$ penalty (e.g., Ravikumar et al., 2009), largely due to the fact that we are under a PAC learning framework and hence there is some discrepancy embedded in the setting in question—that we consider a mis-specified setting without assuming that we know the functional class of the underlying true data generating process, and thereby the tools adopted.

- An important implication of this result is that, in practice, when there is clear separation in magnitude between strong and weak edges, one can effectively obtain a sparse graph by thresholding. This approach practically treats the weak edges as zero, thereby enhancing interpretability.

*Remark* 9 (Refinement of Corollary 2). By incorporating the analysis on the functional approximation error, the above corollary can be refined as follows, under the same setting as considered in Theorem 2:

$$\mathbb{E}_{\boldsymbol{z}}\Bigg[\sum_{j=1}^p \sum_{k \neq j} \mathbb{1}\big\{|\widehat{\beta}_{jk}(\boldsymbol{z})| \geq \tau\big\} \neq \mathbb{1}\big\{|\beta^*_{jk}(\boldsymbol{z})| \geq \underline{\beta}\big\}\Bigg] \lesssim \frac{pL\xi}{\alpha\gamma \min\{\eta^2, (1-\eta)^2\}(\underline{\beta} - \bar{\beta})^2}.$$

## 4 Synthetic Data Experiments

The performance of the proposed neural network-based method is assessed through a series of experiments on synthetic data sets. Both Gaussian and non-Gaussian settings are considered, with samples (indexed by $i$) generated according to one of the following three mechanisms:

- Gaussian: $\boldsymbol{x}^i \sim \mathcal{N}(0, (\Theta^i)^{-1})$.
- Non-paranormal (NPN, Liu et al. (2009)): $\boldsymbol{x}^i = f^{-1}(\check{\boldsymbol{x}}^i)$; $\check{\boldsymbol{x}}^i \sim \mathcal{N}(0, (\Theta^i)^{-1})$ with $f$ being a monotone and differentiable function, and its inverse $f^{-1}$ is applied to $\check{\boldsymbol{x}}^i$ in a coordinate-wise fashion.

- Directed Acyclic Graph (DAG): for each coordinate $j$, $x_j^i = \sum_{k \in \text{pa}(j)} f_{jk}^i(x_k^i) + \epsilon_j$, where $\text{pa}(j)$ is the *parent* set of node $j$. In other words, $\boldsymbol{x}^i$ is generated according to a structural equation model (SEM). Note that the SEM serves solely as a data generation mechanism to introduce potentially nonlinear dependencies (e.g., by parameterizing $f_{jk}$'s through basis functions) amongst the nodes in a flexible way. However, the primary quantity of interest is still the conditional independence structure captured in the corresponding undirected moralized graph, as described next.

For the Gaussian and the non-paranormal cases, $\Theta^i$ is determined by $\boldsymbol{z}^i$. For the DAG case, we start with a binary $A^i$ determined by $\boldsymbol{z}^i$ that corresponds to the skeleton of the DAG (denoted by $G_{\text{DAG}}^i$); the parent set of node $j$ satisfies $\text{pa}(j) \equiv \{k : A_{jk}^i \neq 0\}$. The undirected conditional independence graph of interest that is compatible with the data can then be obtained by *moralizing* the DAG (Cowell et al., 1999, Chapter 3.2.1), namely, $G^i = \mathcal{M}(G_{\text{DAG}}^i)$, whose graph structure is encoded in $\Theta^i$. It can be viewed as the counterpart to those in the Gaussian/non-paranormal case in terms of capturing conditional independence relationships[7]. For all three cases, $\Theta^i$ is the parameter of interest, and we are interested in how well its *skeleton* can be recovered by the proposed method. The results are benchmarked against various competitors, including RegGMM (Zhang & Li, 2023), glasso (Friedman et al., 2008) and nodewise Lasso (Meinshausen & Bühlmann, 2006). RegGMM accounts for covariate dependence while assuming linearity, whereas the other two are graphical model estimation methods that solely consume $\boldsymbol{x}$ as the input.

**Settings.** The description of the various simulation settings is outlined in Table 2. For settings where data are generated according to the Gaussian or NPN models, we first generate "candidate" precision matrices denoted by $\Psi_l$'s, where $\Psi_l$ either has a single band $l$ steps from the main diagonal (for settings G1/N1) or is block diagonal with nonzero entries on the $l$-th block (for settings G2/N2). Next, each $\Theta^i$ is a *convex combination* — which automatically ensures the positive definiteness of $\Theta^i$ — of the candidate precision matrices, and the mixing depends on the value of the covariate $\boldsymbol{z}^i$. For settings D1 and D2 where data are generated according to a DAG, we first generate binary matrices $B_1$ and $B_2$; both $B_1$ and $B_2$ correspond to the skeleton of some trees, with nodes having 1 to 3 children (randomly determined as the tree grows). The covariate $\boldsymbol{z}^i$ dictates the skeleton of DAG, i.e., $A^i$, and also governs $f_{jk}^i$ either directly through the coefficients on the $x_k$'s (linear case), or through a multiplier that impacts the coefficients of the basis functions (non-linear case). For all settings, the samples effectively fall under different "clusters" according to their $\boldsymbol{z}^i$'s. Samples within the same cluster have identical skeletons, albeit the magnitude of the entries may differ depending on the exact value of the $\boldsymbol{z}^i$'s.

The degree of linearity in $\boldsymbol{z}$ varies by setting. For settings G1 and N1, the dependency on $\boldsymbol{z} \in \mathbb{R}^2$ is linear in its 1st coordinate which governs the mixing percentage, and the 2nd coordinate effectively dictates the cluster membership. This is similar for settings D1 and D2, which however differ in that the dependency on the 1st coordinate being quadratic. For settings G2 and N2, the dimension of $\boldsymbol{z}^i$ is set at 10; to induce non-linearity, we use a radial basis function (RBF) network $\varphi$ that first transforms $\boldsymbol{z}$ into a scalar, followed by a sigmoid transformation, namely $\tilde{z}^i := \text{sigmoid}(\varphi(\boldsymbol{z}^i))$ where $\varphi(\boldsymbol{z}) := \sum_{\ell=1}^{L} \alpha_\ell \exp(-\beta_\ell \|\boldsymbol{z} - \boldsymbol{c}_\ell\|^2)$; the sigmoid transformation ensures that $\tilde{z}^i \in (0, 1)$, which then dictates the mixing percentage. Finally, note that amongst these settings, the dependency on the $x_k$'s are linear in settings G1, G2 and D1. Additional details and pictorial illustrations are deferred to Appendix B.

For all settings, we train the model with $\boldsymbol{\beta}(\cdot)$ parameterized with an MLP[8] on 10,000 samples, and evaluate it on a test set of size 1000; the estimated graphs (off-diagonals only) for these test samples (indexed by $i$) are extracted as $\{-\widehat{\beta}_{jk}(\boldsymbol{z}^i)\}_{j,k \in [p]; j \neq k}$. For benchmarking methods that are not covariate-dependent, i.e., glasso and nodewise Lasso, their sample-level estimates are identical, namely $\widehat{\Theta}^i \equiv \widehat{\Theta}$ for all $i$. Instead of directly running the method on the full set of training samples, we further partition them based on their

---

[7]In the special case where $\boldsymbol{x}$ is generated according to a linear SEM, namely $\boldsymbol{x} = A\boldsymbol{x} + \boldsymbol{\epsilon}$, $\Theta := \text{cov}(\boldsymbol{x})$ satisfies $\Theta^{-1} \equiv (I - A^\top)^\top \Omega (I - A^\top)^{-1}$ where $\Omega = \text{cov}(\boldsymbol{\epsilon})$; see, e.g., Loh & Bühlmann (2014). In the absence of linearity, the exact magnitude of the entries in $\Theta$ becomes difficult to infer; however, one can still infer its skeleton from that of the DAG by the definition of moralization.

[8]We use stacked linear layers with ReLU activations, and do not leverage any RBF-style layers or sigmoid ones which are used in the data generating process.

Table 2: Overview of simulation settings.

| | MECHANISM | DESCRIPTION |
|---|---|---|
| G1 | Gaussian | $p = 50, q = 2$; $\begin{bmatrix} z_1^i \\ z_2^i \end{bmatrix} \sim \mathsf{Unif}(0,1)$; $\Theta^i = \begin{cases} z_1^i \Psi_1 + (1 - z_1^i)\Psi_2 & \text{if } z_2^i \in (0, 1/3) \\ z_1^i \Psi_2 + (1 - z_1^i)\Psi_3 & \text{if } z_2^i \in (1/3, 2/3) \\ z_1^i \Psi_1 + (1 - z_1^i)\Psi_3 & \text{if } z_2^i \in (2/3, 1) \end{cases}$ . |
| G2 | Gaussian | $p = 90, q = 10$; $\boldsymbol{z}^i \sim$ standard multivariate Gaussian; $\tilde{z}^i := \mathrm{sigmoid}(\varphi(\boldsymbol{z}^i))$; $\varphi(\cdot)$ is an RBF network; $\varphi(\boldsymbol{z}) := \sum_{\ell=1}^{L} \alpha_\ell \exp(-\beta_\ell \|\boldsymbol{z} - \boldsymbol{c}_\ell\|^2)$, $L = 10$, $\alpha_\ell \sim \mathsf{Unif}(-10, 10)$, $\beta_\ell \sim \mathsf{Unif}(0.1, 0.5)$, $\boldsymbol{c}_\ell \sim [\mathsf{Unif}(-1, 1)]^L$; $\Theta^i = \begin{cases} \tilde{z}^i \Psi_1 + (1 - \tilde{z}^i)\Psi_3 & \text{if } \tilde{z}^i > 0.9 \text{ or } \tilde{z}^i < 0.1 \\ \tilde{z}^i \Psi_1 + 0.5\Psi_2 + (0.5 - \tilde{z}^i)\Psi_3 & \text{otherwise} \end{cases}$ . |
| N1 | NPN | $\check{\boldsymbol{x}}^i$'s are generated identically to those in G1; $\boldsymbol{x}^i = g(\check{\boldsymbol{x}}^i)$ where $g(x) = x + \sin(x)$. |
| N2 | NPN | $\check{\boldsymbol{x}}^i$'s are generated identically to those G2; $\boldsymbol{x}^i = g(\check{\boldsymbol{x}}^i)$ where $g(x) = x^2 \mathrm{sign}(x)$. |
| D1 | DAG | $p = 50, q = 2$; $\begin{bmatrix} z_1^i \\ z_2^i \end{bmatrix} \sim \mathsf{Unif}(-1, 1)$; $\tilde{A}^i = \begin{cases} B_1 & \text{if } z_1^i \in (0, \frac{1}{2}) \\ B_2 & \text{if } z_1^i \in (-\frac{1}{2}, 0) \\ (z_2^i)^2 B_1 + (1 - (z_2^i)^2)B_2 & \text{otherwise} \end{cases}$ ; $f_{jk}^i(x) := \tilde{A}_{jk}^i x_k$; $A^i := \mathbf{1}(\tilde{A}^i \neq 0)$. |
| D2 | DAG | $p, q$, $\boldsymbol{z}^i$'s and $\tilde{A}^i$'s are set identically to those in D1; $f_{jk}^i(x) := \alpha_{jk,1}^i \psi_1(x) + \alpha_{jk,2}^i \psi_2(x) + \alpha_{jk,3}^i \psi_3(x)$ $\psi_m(\cdot)$'s are Gauss-Hermite functions (Olver et al., 2010); $\alpha_{jk,m}^i = (\tilde{A}_{k,j}^i \cdot c_{jk,m})$, $c_{jk,m} \sim \mathsf{Unif}(0.1, 0.5)$. |

cluster membership, and conduct separate estimations using only within-cluster samples.[9] Note that such partition would not be feasible in real world settings as the cluster membership would not be known apriori. All experiments are repeated over 5 data replicates.

**Performance evaluation.** We use AUROC and AUPRC as metrics, which adequately capture how well the methods estimate entries with strong signals versus weak ones[10], and there is no need to apply hard-thresholding to estimates to obtain these two metrics. For a single experiment based on (any) one data replicate, the metrics are initially obtained at the *sample*-level by comparing $\widehat{\Theta}^i$ against $\Theta^i$, for all samples under consideration[11]; to obtain the metrics of interest corresponding to a single experiment, we average those over the samples. For glasso/nodewise Lasso, experiments are conducted over a sequence of penalty parameters, and the highest metric values are reported. Table 3 reports these metrics, after averaging across the data replicates (experiments), with the corresponding standard deviation reported in parentheses.

Table 3: Evaluation for the proposed and benchmarking methods, averaged over 5 data replicates

| | **DNN-based CGM** | | RegGMM | | glasso - est. by cluster | | nodewise Lasso - est. by cluster | |
|---|---|---|---|---|---|---|---|---|
| | AUROC | AUPRC | AUROC | AUPRC | AUROC | AUPRC | AUROC | AUPRC |
| G1 | 0.99 (0.000) | 0.98 (0.001) | 0.96 (0.002) | 0.79 (0.004) | 0.97 (0.000) | 0.59 (0.000) | 0.99 (0.000) | 0.98 (0.002) |
| G2 | 0.99 (0.003) | 0.92 (0.024) | 0.86 (0.017) | 0.47 (0.020) | 0.97 (0.000) | 0.45 (0.031) | 0.96 (0.007) | 0.55 (0.141) |
| N1 | 0.99 (0.000) | 0.98 (0.001) | 0.99 (0.001) | 0.89 (0.001) | 0.98 (0.000) | 0.59 (0.000) | 0.99 (0.001) | 0.98 (0.003) |
| N2 | 0.99 (0.004) | 0.90 (0.028) | 0.82 (0.015) | 0.37 (0.023) | 0.96 (0.000) | 0.42 (0.034) | 0.93 (0.016) | 0.37 (0.164) |
| D1 | 0.95 (0.004) | 0.76 (0.026) | 0.94 (0.005) | 0.74 (0.026) | 0.92 (0.005) | 0.46 (0.008) | 0.94 (0.005) | 0.85 (0.032) |
| D2 | 0.92 (0.008) | 0.66 (0.022) | 0.87 (0.013) | 0.60 (0.017) | 0.89 (0.009) | 0.40 (0.017) | 0.87 (0.015) | 0.50 (0.113) |

The main observations are: (1) the proposed method exhibits superior performance than existing benchmarks, and particularly so in AUPRC. As a matter of fact, all methods exhibit reasonable performance in AUROC, even for glasso/nodewise Lasso that cannot perform sample-specific estimation. Notably, when the dependency on $\boldsymbol{z}$ is linear within each cluster (settings G1 and N1), nodewise Lasso performs almost perfectly when separate estimation is conducted within each cluster and thus on samples whose underlying

---

[9]Recall that that the true graphs corresponding to samples within the same cluster have identical skeletons, although the magnitude may still vary. glasso/nodewise Lasso are still "mis-specified", however to a lesser extent when compared against the case where this "cluster membership" information is ignored.

[10]The estimated $\widehat{\beta}$ does not correspond to edge probabilities, but rather, it corresponds to edge weights. AUROC/AUPRC can effectively summarize how well the graph skeleton is being captured after the entries are thresholded over a range of thresholding levels.

[11]For methods that can produce sample-specific estimates, the evaluation is done on the test set; for those that can only produce a single estimate based on all samples, the evaluation is done on the graph estimated based on the training samples directly.

true graphs have the same skeleton yet different magnitudes, despite that the model assumes that all samples have identical graphs. (2) In the case where the true data generating process is mildly non-linear in $x_k$'s—in particular, the non-linearity is induced via monotonic transformations (i.e., N1 and N2), the proposed formulation in (2) as a working model—by only considering linear dependency on the $x_k$'s—is still able to recover well the skeleton. (3) In the case where the true DGP becomes highly non-linear (e.g., D2), the performance of the method deteriorates. This is expected in that the moral graph can be inherently difficult to recover, partly due to the existence of edges that are induced by "married" nodes and have small magnitude; the non-linearity in $x_k$'s can further add to the complexity of edge recovery. The former is corroborated by D1, where although the dependency on $x_k$'s is linear, the skeleton recovery is still inferior. Additional tables and remarks are deferred to Appendix B for further benchmarking and illustration, including the skeleton recovery (0/1) performance of DNN-CGM at different thresholding levels (Table 4), the performance for the DAG setting when estimates are evaluated against the "pseudo" moralized graph without the edges from the married nodes (Table 5), and the performance of glasso/nodewise Lasso without sample partition (Table 6).

## 5 Real Data Experiments

To demonstrate how the proposed method performs in real world settings, we consider two applications from the domains of neuroscience and finance (deferred to Appendix C), and assess the network structures recovered.

We consider a dataset from the Human Connectome Project analyzed in Lee et al. (2023) comprising of resting-state fMRI scans for 549 subjects. The scan of any subject is in the form of a spatial-temporal matrix, with the rows corresponding to the "snapshot" value of 268 brain regions and the columns being the temporal observations of the region, totaling 1200 time points. In addition, each subject is associated with a Penn Progressive Matrices score, a surrogate of fluid intelligence. The quantity of interest is the network of brain regions ($p = 268$) as a function of the intelligence score ($q = 1$). The dataset has been pre-processed with global signal regression, where shared variance between the global signal and the time course of each individual voxel is removed through linear regression (Murphy & Fox, 2017; Greene et al., 2018). In our experiment, we flatten the time dimension of the fMRI scans and ignore the temporal nature of these scans; therefore, the dataset contains $1200 \times 549 \approx 660$k effective samples. However, note that since the score is at the subject level, the estimation procedure yields a single unique network for each subject[12].

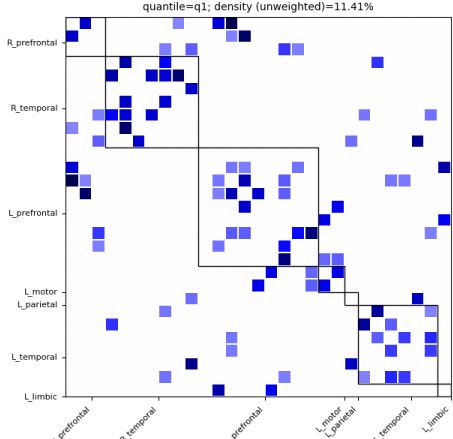

(a) Network for subjects whose scores are in the 1st quantile

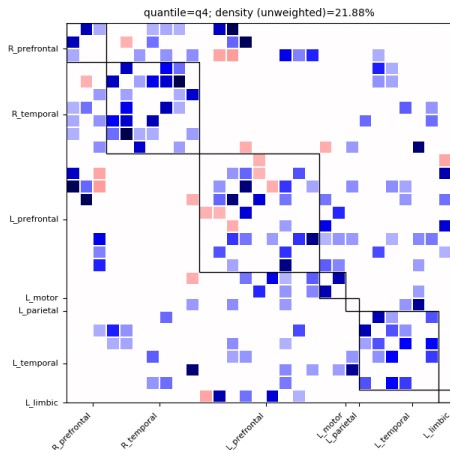

(b) Network for subjects whose scores are in the 4th quantile

Figure 1: Estimated networks as represented by $-\widehat{\beta}(z)$ for subjects having low (left) and high scores (right), after thresholding at 0.05. Red cells indicate positive partial correlations and blue cells indicate negative ones.

---

[12]This is due to the fact that the model is specified as a function of the score and thus effective samples of the same subject will always have their sample-specific estimate graphs being identical.

**Results.** We group the subjects based on the quantiles that their respective scores fall into, and obtain the averaged graph over subjects that are within the same quantile band (labeled as q1, q2, q3 and q4). Further, for visualization purposes, we partition the nodes into 6 sub-networks, corresponding to `default mode`, `frontoparietal`, `medial frontal`, `motor`, `subcortical cerebellar` and `visual`, as in Lee et al. (2023). Figure 1 shows the comparison between the medial frontal sub-networks of subjects in q1 and q4, respectively, after the estimated graphs are sparsified (see Remark 3) and thresholded at 0.05 (on a normalized scale). Note that each node is associated with a label[13]; in the plot, nodes with the same label are grouped together.

The difference in the network density is rather pronounced; in particular, those having a higher score exhibit a significantly more connected network. Similar patterns are also observed in other sub-networks; in particular, the differential between q1 and q4 networks of `frontoparietal`, `motor` and `visual` is of comparable scale to that in `medial frontal`, while that of `default mode` and `subcortical-cerebellar` is less pronounced. This is largely concordant with observations from existing literature (Song et al., 2008; Ohtani et al., 2014) and is also corroborated in a validation dataset encompassing 828 subjects; the estimated graphs therein exhibit the same pattern.

## 6 Discussion

A nonlinear covariate-dependent graphical model based on a node-conditional formulation is investigated. The functional dependency on the covariate $z$ is parameterized by neural networks, and hence the model can capture this dependency in a flexible way. Theoretical guarantees are provided under the PAC learning framework, wherein both the generalization error and the approximation error are taken into account.

At the methodological level, alternative approaches similar to the formulation in Baptista et al. (2024b), can be potentially adopted, by considering a modified score matrix $\Omega$ that takes into account the external covariate, namely $\Omega_{jk} := \mathbb{E}_{p(\boldsymbol{x}|\boldsymbol{z})}[\partial_j \partial_k \log p(\boldsymbol{x}|\boldsymbol{z})]^2$. Consequently, estimation can proceed with the use of a lower triangular transport map to estimate the conditional density $p(\boldsymbol{x}|\boldsymbol{z})$ (Baptista et al., 2024a), or score matching to approximate the conditional score $\nabla_{\boldsymbol{x}} \log p(\boldsymbol{x}|\boldsymbol{z})$ (Dasgupta et al., 2023). However, it is worth noting that the consistency result established in Baptista et al. (2024b) is under an asymptotic regime, where the number of nodes $p$ is assumed fixed and the sample size $n$ grows. Their analysis relies on a Taylor expansion, followed by steps that invoke the delta method and the continuous mapping theorem, under some additional assumptions[14]. In contrast, the theoretical result in this paper establishes a finite-sample error bound for the estimator under a regime where $p$ can grow slowly with $n$. Hence, the analysis requires a different set of technical tools, even in the absence of the covariate $z$. By adotping a node-conditional formulation, the learning task reduces to an ERM problem, enabling us to leverage existing results in the supervised learning literature.

Finally, the focus of the paper is on graphical models for continuous variables $\boldsymbol{x}$, but with appropriate modifications the proposed framework can be extended to discrete random variables—e.g., a covariate dependent Ising and/or Potts model (Wainwright et al., 2008)—with a cross-entropy loss function. Theoretical results can potentially be established using a similar set of arguments, leveraging results in ERM for the case of classification.

### Acknowledgements

The authors thank the Action Editor and three anonymous reviewers for their careful review of the work, and their constructive comments and suggestions.

George Michailidis was supported in part by NSF grants DMS-2348640 and DMS-2319593.

---

[13]Amongst all 268 nodes, there are 20 distinct labels

[14]Two key assumptions include: (i) the parameterization—based on bases expansions—of the transport map is sufficiently rich to cover the target density, and (ii) the estimated $\Omega$ before thresholding is an exact expectation, namely, $\widehat{\Omega}_{jk} := \mathbb{E}[\partial_j \partial_k \log \widehat{p}(\boldsymbol{x})]^2$, rather than sample-level estimates.

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

# A    Technical Appendix

## A.1    Technical Definitions and A Useful Lemma

For notational convenience, the risk function is abbreviated to $\mathcal{L}(\boldsymbol{\beta}) := \frac{1}{p}\sum_{j=1}^{p}\ell\left(\sum_{k\neq j}\beta_{jk}(\boldsymbol{z})x_k, x_j\right)$ for the remainder of the presentation. Let $\mathcal{H} := \{h : \mathcal{Y} \to \mathbb{R}\}$ be a family of measurable functions, and $\{\sigma^i\}_{i=1}^n \in \{-1, +1\}^n$ a collection of i.i.d Rademacher random variables. The *Rademacher Complexity* and *Rademacher Average* (Bartlett & Mendelson, 2002) are defined as:

$$\mathfrak{R}_n\mathcal{H} = \mathbb{E}_{\{\sigma^i\}_{i=1}^n}\left[\sup_{h\in\mathcal{H}}\frac{1}{n}\sum_{i=1}^n\sigma^i h(\boldsymbol{y}^i)\right], \qquad \mathfrak{R}\mathcal{H} = \mathbb{E}_{\boldsymbol{y}_{1:n},\{\sigma^i\}_{i=1}^n}\left[\sup_{h\in\mathcal{H}}\frac{1}{n}\sum_{i=1}^n\sigma^i h(\boldsymbol{y}^i)\right].$$

Further, we define the empirical average and population expectation of $h$, as

$$\mathbb{P}_n h := \frac{1}{n}\sum_{i=1}^n h(\boldsymbol{y}) \qquad \text{and} \qquad \mathbb{P}h := \mathbb{E}_{\boldsymbol{y}}h(\boldsymbol{y}).$$

In particular, we view $\boldsymbol{y} := (\boldsymbol{x}, \boldsymbol{z})$ and $h(\boldsymbol{y}) := (\ell \circ \boldsymbol{\beta})(\boldsymbol{y}) := \mathcal{L}(\boldsymbol{\beta}; \boldsymbol{x}, \boldsymbol{z})$ throughout our analysis. We further define the *Local Rademacher Complexity* and *Local Rademacher Average* with radius $r$ as $\mathfrak{R}_n\{h \in \mathcal{H}, \mathbb{P}_n h^2 \leq r\}$ and $\mathfrak{R}\{h \in \mathcal{H}, \mathbb{P}h^2 \leq r\}$. The star hull of set of functions $\mathcal{F}$ is defined as

$$*\mathcal{F} := \{\alpha f : f \in \mathcal{F}, \alpha \in [0, 1]\}.$$

The following concepts are used in the proofs of the main results:

**Definition 1** ($L_2$-Covering Number). Let $S_n := \{\boldsymbol{y}^i\}_{i=1}^n$ be a set of points with $\boldsymbol{y}^i \in \mathcal{Y}, \forall i$. A set $\mathcal{U} \subseteq \mathbb{R}^n$ is an $\varepsilon$-cover w.r.t $L_2$-norm of $\mathcal{F}$ on $\{\boldsymbol{y}^i\}_{i=1}^n$, if $\forall \beta \in \mathcal{F}, \exists \boldsymbol{u} \in \mathcal{U}$, s.t. $\sqrt{\frac{1}{n}\sum_{i=1}^n |u_i - \beta(\boldsymbol{y}^i)|^2} \leq \varepsilon$, where $u_i$ is the $i$-th coordinate of $\boldsymbol{u}$. The covering number $\mathcal{N}_2(\varepsilon, \mathcal{F}, S_n)$ with $L_2$-norm of $\mathcal{F}$ on $S_n$ is :

$$\min\{|\mathcal{U}|: \mathcal{U} \text{ is an } \varepsilon\text{-cover of } \mathcal{F} \text{ on } S_n\},$$

and the covering number of $\mathcal{F}$ with $L_2$-norm of size $n$ is $\mathcal{N}_2(\varepsilon, \mathcal{F}, n) := \sup_{S_n \in \mathcal{Y}^n} \mathcal{N}_2(\varepsilon, \mathcal{F}, S_n)$

**Definition 2** (VC-dimension (Vapnik & Chervonenkis, 1971)). The VC-dimension $d_{\mathrm{VC}}(\mathcal{H})$ of a hypothesis class $\mathcal{H} = \{h : \mathcal{Y} \mapsto \{1, -1\}\}$ is the largest cardinality of the set $S \subseteq \mathcal{Y}$ such that for all subsets of $S$, denoted by $\bar{S}, \exists h \in \mathcal{H}$:

$$f(\boldsymbol{y}) = \begin{cases} 1 & \text{if } \boldsymbol{y} \in \bar{S}, \\ -1 & \text{if } \boldsymbol{y} \in S \setminus \bar{S}. \end{cases}$$

**Definition 3** (Pseudo-dimension (Pollard, 1990)). The Pseudo-dimension $d_P(\mathcal{H})$ of a *real-valued* hypothesis class $\mathcal{H} = \{h : \mathcal{Y} \mapsto [a, b]\}$ is the VC-dimension of the hypothesis class

$$\tilde{\mathcal{H}} = \{\tilde{h} : \mathcal{Y} \times \mathbb{R} \mapsto \{-1, 1\} \mid \tilde{h}(\boldsymbol{y}, t) = \mathrm{sign}(h(\boldsymbol{y}) - t), h \in \mathcal{H}\}.$$

**Definition 4** ($(m, C)$-smoothness (Kohler & Langer, 2021)). Let $m := t + s$ for some $t \in \mathbb{N}_0$ and $0 < s \leq 1$. A function $f(\cdot) : \mathbb{R}^q \mapsto \mathbb{R}$ is called $(m, C)$-*smooth* if for every $\alpha = (\alpha_1, \alpha_2, ..., \alpha_q)$ with $\sum_{j=1}^q \alpha_j = t$, the partial derivative $\partial^t f/(\partial^{\alpha_1} z_1, ..., \partial^{\alpha_q} z_q)(\boldsymbol{z})$ exists and satisfies:

$$\left|\frac{\partial^t f}{(\partial^{\alpha_1} z_1, ..., \partial^{\alpha_d} z_q)}(\boldsymbol{z}) - \frac{\partial^t f}{(\partial^{\alpha_1} z_1, ..., \partial^{\alpha_d} z_q)}(\boldsymbol{y})\right| \leq C\|\boldsymbol{z} - \boldsymbol{y}\|^s.$$

The following Lemma is established which will be used in the proof of Theorem 1.

**Lemma 1** (Verification of the Bernstein Condition). Under Assumptions 1 and 2, the following inequality holds:

$$\mathbb{E}_{\boldsymbol{x},\boldsymbol{z}}[(\mathcal{L}(\widehat{\boldsymbol{\beta}}) - \mathcal{L}(\boldsymbol{\beta}))^2] \leq \frac{L^2}{2\alpha}\mathbb{E}_{\boldsymbol{x},\boldsymbol{z}}\left[\mathcal{L}(\widehat{\boldsymbol{\beta}}) - \mathcal{L}(\boldsymbol{\beta})\right].$$

*Proof.*

$$\mathbb{E}_{\boldsymbol{x},\boldsymbol{z}}[(\mathcal{L}(\widehat{\boldsymbol{\beta}}) - \mathcal{L}(\boldsymbol{\beta}))^2] \leq \mathbb{E}_{\boldsymbol{x},\boldsymbol{z}}\left[L^2 \sum_{j=1}^{p} |\sum_{k\neq j} \beta_{jk}(\boldsymbol{z})x_k - \sum_{k\neq j} \beta_{jk}^*(\boldsymbol{z})x_k|^2\right]$$

$$\leq \frac{L^2}{2\alpha} \sum_{j=1}^{p} \mathbb{E}_{\boldsymbol{x},\boldsymbol{z}}\left[\ell\left(\sum_{k\neq j} \beta_{jk}(\boldsymbol{z})x_k, x_j\right) - \ell\left(\sum_{k\neq j} \beta_{jk}^*(\boldsymbol{z})x_k, x_j\right)\right]$$

$$= \frac{L^2}{2\alpha} \mathbb{E}_{\boldsymbol{x},\boldsymbol{z}}\left[\mathcal{L}(\widehat{\boldsymbol{\beta}}) - \mathcal{L}(\boldsymbol{\beta})\right].$$

$\square$

## A.2 Verification of Assumptions 1 and 2 For the Mean Squared Error Loss

Note that the mean squared error (MSE) loss function (i.e., $\ell(a,b) := (a-b)^2$) is used to recover the structure of the graphical model (see (3)). Next, we establish that it satisfies Assumptions 1 and 2.

Recall that we assumed that the $x_j$'s are uniformly bounded; e.g., without loss of generality we consider $\|\boldsymbol{x}\|_2 \leq 1$. Further, let $\beta_{jk}(\cdot) \in \mathcal{F}$ and $\mathcal{F}$ belongs to family of uniformly bounded functions: $\|\boldsymbol{\beta}(\boldsymbol{z})\|_2 \leq 1$.

Next, we establish that the MSE loss is 1-strongly convex and 4-Lipschitz. Note that being a quadratic function it is trivially 1-strongly convex. The Lipschitz continuity claim follows from:

$$\ell(\widehat{x}_1, x) - \ell(\widehat{x}_2, x) = (\widehat{x}_1 - \widehat{x}_2)(\widehat{x}_2 + \widehat{x}_1 - 2x) \leq 4|\widehat{x}_1 - \widehat{x}_2|.$$

Next, we show that Assumption 2 holds; namely, $\mathbb{E}_{\boldsymbol{x},\boldsymbol{z}}[\sum_{j=1}^{p} \ell'(\langle\boldsymbol{\beta}_j, \boldsymbol{x}_{-j}\rangle, x_j)_{|\beta_j=\beta_j^*}] = \mathbb{0}$. Recall that the posited DGP is:

$$x_j = \sum_{k\neq j}^{p} \beta_{jk}^*(\boldsymbol{z})x_j + \varepsilon_j.$$

Then, the following calculation shows that Assumption 2 holds:

$$\mathbb{E}_{\boldsymbol{x},\boldsymbol{z}}\left[\sum_{j=1}^{p} \ell'(\langle\boldsymbol{\beta}_j, \boldsymbol{x}_{-j}\rangle, x_j)\right] = \mathbb{E}_{\boldsymbol{x},\boldsymbol{z}}\left[\sum_{j=1}^{p}(\langle\boldsymbol{\beta}_j, \boldsymbol{x}_{-j}\rangle - \langle\boldsymbol{\beta}_j^*, \boldsymbol{x}_{-j}\rangle - \varepsilon_j)\right]$$

$$= \underbrace{\mathbb{E}_{\boldsymbol{x},\boldsymbol{z}}\left[\sum_{j=1}^{p}(\langle\boldsymbol{\beta}_j - \boldsymbol{\beta}_j^*, \boldsymbol{x}_{-j}\rangle)\right]}_{=0 \text{ if } \beta_j = \beta_j^*}.$$

## A.3 Proof of Theorem 1

For ease and conciseness of the presentation, the hypothesis class $\mathcal{H}$ and its members are defined as the excess risk, the main quantity of interest in this analysis.

*Proof.* We start by defining the following function class:

$$\mathcal{H} := \Delta \circ \mathcal{L} \circ \mathcal{F} := \left\{\Delta\mathcal{L}(\boldsymbol{\beta}; \boldsymbol{\beta}^*, \boldsymbol{x}, \boldsymbol{z}) = \mathcal{L}(\boldsymbol{\beta}) - \mathcal{L}(\boldsymbol{\beta}^*) : \beta \in \mathcal{F}_{p\times(p-1)}\right\}.$$

Correspondingly, we denote $h(\boldsymbol{x}, \boldsymbol{z}) := \Delta_{\mathcal{L},\boldsymbol{\beta}} := \Delta\mathcal{L}(\boldsymbol{\beta}; \boldsymbol{\beta}^*, \boldsymbol{x}, \boldsymbol{z})$, which is a composite function of $\boldsymbol{\beta}$ from $\mathcal{F}_{p\times(p-1)}$, loss function $\mathcal{L}$ and the difference operation $\Delta$. It could be easily verified that $\forall h \in \mathcal{H}, h(\boldsymbol{x}, \boldsymbol{z}) \in [0,1]$. In addition, $h(\cdot)$ also satisfies the Bernstein Condition by Lemma 1. On the other hand, directly bounding the Pseudo-dimension of $\mathcal{H}$ is not trivial. To that end, we leverage the Lipschitz property of $\ell$

and the covering number of $\mathcal{F}$ to construct an upper-bound on the covering number of $\mathcal{H}$. We will focus on analyzing the Local Rademacher Average $\mathbb{E}\mathfrak{R}_n\{h \in \mathcal{H} : \mathbb{P}_n h^2 \leq r\}$:

$$\mathbb{E}\mathfrak{R}_n\{h \in \mathcal{H}, \mathbb{P}_n h^2 \leq r\} = \mathbb{E}_{S_n \{\sigma^i\}_{i=1}^n} \left[ \sup_{h \in \mathcal{H}, \mathbb{P}_n h^2 \leq r} \frac{1}{n} \sum_{i=1}^n \sigma^i h(\boldsymbol{x}^i, \boldsymbol{z}^i) \right] = \mathbb{E}_{S_n \{\sigma^i\}_{i=1}^n} \left[ \sup_{\substack{\boldsymbol{\beta} \in \mathcal{F}_{p \times (p-1)} \\ \mathbb{P}_n \Delta^2_{\mathcal{L}, \widehat{\boldsymbol{\beta}}} \leq r}} \frac{1}{n} \sum_{i=1}^n \sigma^i \Delta_{\mathcal{L}, \widehat{\boldsymbol{\beta}}} \right].$$

Next, we bound $\mathbb{P}\Delta_{\mathcal{L}, \widehat{\boldsymbol{\beta}}} - \mathbb{P}_n \Delta_{\mathcal{L}, \widehat{\boldsymbol{\beta}}}$. To invoke Theorem 3.3 in Bartlett et al. (2005), we need to find a subroot function $\tau(r)$ such that

$$\tau(r) \geq \frac{2L^2}{\alpha} \mathbb{E}\mathfrak{R}_n\{h \in \mathcal{H} : \mathbb{E}[h^2] \leq r\}.$$

By Lemma 3.4 from Bartlett et al. (2005), it suffices to choose:

$$\tau(r^*) = \frac{20L^2}{\alpha} \mathbb{E}\mathfrak{R}_n\{*\mathcal{H}, \mathbb{P}h^2 \leq r\} + \frac{11 \log n}{n},$$

with fixed point $r^* = \tau(r^*)$ denoted as $r^*$.

The following analysis largely follows from the proof in Corollary 3.7 in Bartlett et al. (2005). Since $\Delta_{\mathcal{L}, \widehat{\boldsymbol{\beta}}}$ is uniformly bounded by 1, for any $r \geq \tau(r)$, Corollary 2.2 in Bartlett et al. (2005) implies that with probability at least $1 - \frac{1}{n}$, $\{h \in *\mathcal{H} : \mathbb{P}h^2 \leq r\} \subseteq \{h \in *\mathcal{H} : \mathbb{P}_n h^2 \leq 2r\}$. Let $\mathcal{E} := \{h \in *\mathcal{H} : \mathbb{P}h^2 \leq r\} \subseteq \{h \in *\mathcal{H} : \mathbb{P}_n h^2 \leq 2r\}$, then the following holds:

$$\mathbb{E}\mathfrak{R}_n\{*\mathcal{H}, \mathbb{P}h^2 \leq r\} \leq \mathbb{P}[\mathcal{E}]\mathbb{E}[\mathfrak{R}_n\{*\mathcal{H}, \mathbb{P}h^2 \leq r\}|\mathcal{E}] + \mathbb{P}[\mathcal{E}^c]\mathbb{E}[\mathfrak{R}_n\{*\mathcal{H}, \mathbb{P}h^2 \leq r\}|\mathcal{E}^c]$$

$$\leq \mathbb{E}[\mathfrak{R}_n\{*\mathcal{H}, \mathbb{P}_n h^2 \leq 2r\}] + \frac{1}{n}.$$

Since $r^*$ is the fixed point of a sub-root function, namely $r^* = \tau(r^*)$, by Lemma 3.2 in Bartlett et al. (2005), $r^*$ satisfies the following

$$r^* \leq \frac{20L^2}{\alpha} \mathbb{E}\mathfrak{R}_n\{*\mathcal{H}, \mathbb{P}_n h^2 \leq 2r^*\} + \frac{11 \log n + 20}{n}, \tag{10}$$

where the Lipschitz constant $L$ and the strong convexity parameter $\alpha$ show due to the Bernstein condition.

Next, we leverage Dudley's chaining bound (Dudley, 2016) to upper bound $\mathbb{E}\mathfrak{R}_n\{*\mathcal{H}, \mathbb{P}_n h^2 \leq 2r^*\}$, using the integral of covering number. Specifically, by applying the chaining bound, it follows from Theorem B.7 (Bartlett et al., 2005) that

$$\mathbb{E}_{S_n}[\mathfrak{R}_n(*\mathcal{H}, \mathbb{P}_n h^2 \leq 2r^*)] \leq \frac{\text{const}}{\sqrt{n}} \mathbb{E}_{S_n} \int_0^{\sqrt{2r^*}} \sqrt{\log \mathcal{N}_2(\varepsilon, *\mathcal{H}, S_n)} d\varepsilon, \tag{11}$$

where const represents some universal constant. Next, we bound the covering number $\mathcal{N}_2(\varepsilon, *\mathcal{H}, S_n)$ by $\mathcal{N}_2(\varepsilon/Lp, \mathcal{F}_{p \times (p-1)}, S_n)$. We show that for all $S_n$, any $\frac{\varepsilon}{Lp}$-cover of $\mathcal{F}_{p \times (p-1)}$ is a $\varepsilon$-cover of $\mathcal{H}$, which implies that $\mathcal{N}_2(\varepsilon, *\mathcal{H}, S_n) \leq \mathcal{N}_2(\varepsilon/Lp, \mathcal{F}_{p \times (p-1)}, S_n)$. Specifically, let $\mathcal{U}_{jk} \subset [0, 1]^n$ be an $\varepsilon$-cover of $\mathcal{F}$ on $S_n$ so that for all $\beta_{jk} \in \mathcal{F}$, $\exists \{u^i_{jk}\}_{i=1}^n \in \mathcal{U}_{jk}$ so that

$$\sqrt{\frac{1}{n} \sum_{i \in [n]} (\beta_{jk}(\boldsymbol{z}^i) - \boldsymbol{u}^i_{jk})^2} \leq \varepsilon.$$

Further, let $\mathcal{U} \subseteq \mathbb{R}^{n \times p \times (p-1)}$, $\boldsymbol{\beta}(\boldsymbol{z}) : \mathcal{Z} \to \mathbb{R}^{p \times (p-1)}$ where $\boldsymbol{\beta}(\boldsymbol{z}) \in \mathcal{F}_{p \times (p-1)}$ is a family of $p \times (p-1)$ joint functions, with each element $\beta_{jk} \in \mathcal{F}, j, k \in [p], j \neq k$. We say $\mathcal{U}$ is an $\varepsilon$-cover of $\mathcal{F}_{p \times (p-1)}$ on $\{\boldsymbol{z}^i\}_{i=1}^n$ if $\forall \boldsymbol{\beta} \in \mathcal{F}_{p \times (p-1)}$, $\exists \boldsymbol{u} \in \mathcal{U}$, s.t.

$$\sqrt{\frac{1}{np(p-1)} \sum_{i \in [n]} \sum_{j \in [p]} \sum_{k \neq j} (u^i_{jk} - \beta_{jk}(\boldsymbol{z}^i))^2_2} \leq \varepsilon.$$

Clearly, let $\mathcal{U}_{jk}$ be any collection of arbitrary $p \times (p-1)$ $\varepsilon$-covers of $\mathcal{F}$, the Cartesian product of $\mathcal{U}_{jk}, j, k \in [p], j \neq k$ forms an $\varepsilon$-cover of $\mathcal{F}_{p \times (p-1)}$, which implies that $|\mathcal{U}| \leq |\mathcal{U}_{jk}|^{p^2}$. Thus we have $\mathcal{N}_2\big(\varepsilon, \mathcal{F}_{p \times (p-1)}, S_n\big) \leq \mathcal{N}_2\big(\varepsilon, \mathcal{F}, S_n\big)^{p^2}$. Next, we show that given any $\frac{\varepsilon}{Lp}$-cover of $\mathcal{F}_{p \times (p-1)}$ on $S_n$, denoted as $\mathcal{U}$, one can construct $\mathcal{V} := \{\boldsymbol{v} = (v^1, \cdots, v^n)' \in \mathbb{R}^n | v^i := \frac{1}{p}\sum_j^p \big(\ell(\langle \boldsymbol{u}_j^i, \boldsymbol{x}_{-j}^i \rangle, x_j^i) - \ell(\langle \boldsymbol{\beta}_j^*, \boldsymbol{x}_{-j}^i \rangle, x_j^i)\big), i \in [n], \boldsymbol{u} \in \mathcal{U}\}$, which is an $\varepsilon$-cover for $\mathcal{H}$, i.e., for all $h \in \mathcal{H}, \exists~\boldsymbol{v} \in \mathcal{V}$ so that $\sqrt{\frac{1}{n}\sum_{i \in [n]}(h(\boldsymbol{x}^i, \boldsymbol{z}^i) - v^i)^2} \leq \varepsilon$ :

$$\sqrt{\frac{1}{n}\sum_{i \in [n]}\left(\frac{1}{p}\sum_{i \in [p]}\ell\big(\langle \boldsymbol{u}_j^i, \boldsymbol{x}_{-j}^i \rangle, x_j\big) - \frac{1}{p}\sum_{j \in [p]}\ell\big(\langle \boldsymbol{\beta}_j^*, \boldsymbol{x}_{-j}^i \rangle, x_j\big) - \Delta_{\mathcal{L},\boldsymbol{\beta}}(\boldsymbol{x}^i, \boldsymbol{z}^i)\right)^2}$$

$$= \sqrt{\frac{1}{n}\sum_{i \in [n]}\left(\frac{1}{p}\sum_{j \in [p]}\ell\big(\langle \boldsymbol{u}_j^i, \boldsymbol{x}_{-j}^i \rangle, x_j\big) - \frac{1}{p}\sum_{j \in [p]}\ell\big(\langle \boldsymbol{\beta}_j, \boldsymbol{x}_{-j}^i \rangle, x_j\big)\right)^2}$$

$$\overset{(1)}{\leq} \sqrt{\frac{1}{np^2}\sum_{k \in [n]}\left(\sum_{i \in [p]}L|\langle \boldsymbol{u}_i^i - \boldsymbol{\beta}_j, \boldsymbol{x}_{-j}^i \rangle|\right)^2}$$

$$\overset{(2)}{\leq} \sqrt{\frac{L^2}{np}\sum_{i \in [n]}\sum_{j \in [p]}\left(|\langle \boldsymbol{u}_j^i - \boldsymbol{\beta}_j, \boldsymbol{x}_{-j}^i \rangle|\right)^2}$$

$$\overset{(3)}{\leq} \varepsilon Lp.$$

In the above derivation, (1) leverages the fact that

$$\Delta_{\mathcal{L},\boldsymbol{\beta}}(\boldsymbol{x}^i, \boldsymbol{z}^i) = \frac{1}{p}\sum_{i \in [p]}\ell\left(\langle \boldsymbol{\beta}_j^i, \boldsymbol{x}_{-j}^i \rangle, x_j\right) - \frac{1}{p}\sum_{j \in [p]}\ell\left(\langle \boldsymbol{\beta}_j^*, \boldsymbol{x}_{-j}^i \rangle, x_j\right);$$

(2) is by Lipschitz continuity, and (3) is by the definition of the covering number and the fact that $|x_k^i| \leq 1, \forall k \in [p]$. Combining the above inequality with Corollary 3.7 from Bartlett et al. (2005), we have

$$\log \mathcal{N}_2(\varepsilon, *\mathcal{H}, S_n) \leq \log\left\{\mathcal{N}_2\Big(\frac{\varepsilon}{2}, \mathcal{H}, S_n\Big)\Big(\lceil\frac{2}{\varepsilon}\rceil + 1\Big)\right\}$$

$$\leq \log\left\{\mathcal{N}_2\Big(\frac{\varepsilon}{8Lp}, \mathcal{F}_{p \times (p-1)}, S_n\Big)\Big(\lceil\frac{2}{\varepsilon}\rceil + 1\Big)\right\}$$

$$\leq p^2 \log\left\{\mathcal{N}_2\Big(\frac{\varepsilon}{8Lp}, \mathcal{F}, S_n\Big)\Big(\lceil\frac{2}{\varepsilon}\rceil + 1\Big)\right\}.$$

Next, we bound $\frac{\text{const}}{\sqrt{n}}\mathbb{E}\int_0^{\sqrt{2r^*}}\sqrt{\log \mathcal{N}_2(\varepsilon, *\mathcal{H}, S_n)}d\varepsilon$ from (11). Note that by Haussler's bound on the covering number (Haussler, 1995) we have:

$$\log \mathcal{N}_2\Big(\frac{\varepsilon}{8Lp}, \mathcal{F}, S_n\Big) \leq \text{const} \cdot d_P(\mathcal{F})\log\Big(\frac{Lp}{\varepsilon}\Big), \quad \forall~S_n,$$

and therefore

$$\frac{\text{const}}{\sqrt{n}}\mathbb{E}_{S_n}\int_0^{\sqrt{2r^*}}\sqrt{\log \mathcal{N}_2(\varepsilon, *\mathcal{H}, S_n)}d\varepsilon \leq \frac{\text{const}}{\sqrt{n}}\mathbb{E}_{S_n}\int_0^{\sqrt{2r^*}}\sqrt{\log \mathcal{N}_2\Big(\frac{\varepsilon}{2}, \mathcal{H}, S_n\Big)\Big(\lceil\frac{2}{\varepsilon}\rceil + 1\Big)}d\varepsilon$$

$$\leq \frac{\text{const} \cdot p}{\sqrt{n}}\mathbb{E}_{S_n}\int_0^{\sqrt{2r^*}}\sqrt{\log \mathcal{N}_2\Big(\frac{\varepsilon}{8Lp}, \mathcal{F}, S_n\Big)\Big(\lceil\frac{2}{\varepsilon}\rceil + 1\Big)}d\varepsilon$$

$$\leq \text{const} \cdot p\sqrt{\frac{d_P(\mathcal{F})}{n}}\int_0^{\sqrt{2r^*}}\sqrt{\log\Big(\frac{Lp}{\varepsilon}\Big)} \leq \text{const} \cdot p\sqrt{\frac{d_P(\mathcal{F})r^* \log(L/r^*)}{n}}$$

$$\leq \text{const} \cdot p\sqrt{\frac{d_P^2(\mathcal{F})}{n^2} + \frac{d_P(\mathcal{F})r^* \log(nLp/ed_P(\mathcal{F}))}{n}}, \tag{12}$$

where $e$ in (12) refers to Euler's constant and const represents some universal constant that may change from line to line in the above derivation. The inequality in (12) comes from the fact that $r^* \cdot \log(1/r^*)$ is a monotone increasing function for $r^* \leq e$: in the case where $r^* \leq \frac{ed}{n} \frac{1}{\log^2(n/ed)} \leq e$, we have $\frac{d_P(\mathcal{F})r^* \log(1/r^*)}{n} \lesssim \frac{d_P^2(\mathcal{F})}{n^2}$; in the case where $r^* \geq \frac{ed}{n} \frac{1}{\log^2(n/ed)}$, we have $\frac{d_P(\mathcal{F})r^* \log(1/r^*)}{n} \lesssim \frac{d_P(\mathcal{F})r^*}{n} \log(n/ed_P(\mathcal{F}))$. Combining these two cases yields inequality (12).

Together with (10) one can solve for

$$r^* \lesssim \frac{p^2 L^4 d_P(\mathcal{F}) \log(\frac{nLp}{d_P(\mathcal{F})})}{\alpha^2 n}.$$

By Theorem 3.3 in Bartlett et al. (2005), we have that for all $\boldsymbol{\beta} \in \mathcal{F}_{p \times (p-1)}$, with probability at least $1 - \delta$:

$$\mathcal{R}(\boldsymbol{\beta}) - \mathcal{R}(\boldsymbol{\beta}^*) \lesssim \mathcal{R}_n(\boldsymbol{\beta}) - \mathcal{R}_n(\boldsymbol{\beta}^*) + \frac{d_P(\mathcal{F})}{\alpha n} L^2 p^2 \log(nLp) \log\left(\frac{1}{\delta}\right),$$

$$\mathcal{R}_n(\boldsymbol{\beta}) - \mathcal{R}_n(\boldsymbol{\beta}^*) \lesssim \mathcal{R}(\boldsymbol{\beta}) - \mathcal{R}(\boldsymbol{\beta}^*) + \frac{d_P(\mathcal{F})}{\alpha n} L^2 p^2 \log(nLp) \log\left(\frac{1}{\delta}\right), \tag{13}$$

which gives

$$\underbrace{\mathcal{R}(\widehat{\boldsymbol{\beta}}) - \mathcal{R}(\boldsymbol{\beta}^*)}_{\text{excess error}} \lesssim \underbrace{\mathcal{R}_n(\widehat{\boldsymbol{\beta}}) - \mathcal{R}_n(\boldsymbol{\beta}^*)}_{\text{empirical excess error}} + \underbrace{\frac{d_P(\mathcal{F})}{\alpha n} L^2 p^2 \log(nLp) \log\left(\frac{1}{\delta}\right)}_{\text{generalization error}}. \tag{14}$$

Due to the fact that $\mathcal{R}(\widehat{\boldsymbol{\beta}})$ is the empirical risk minimizer, we have:

$$\underbrace{\mathcal{R}_n(\widehat{\boldsymbol{\beta}}) - \mathcal{R}_n(\boldsymbol{\beta}^*)}_{\text{empirical excess error}} = \frac{1}{np} \sum_{i \in [n]} \left[ \sum_{j=1}^p \ell\left( \langle \widehat{\boldsymbol{\beta}}_j(\boldsymbol{z}^i), \boldsymbol{x}_{-j}^i \rangle, x_j^i \right) - \sum_{j=1}^p \ell\left( \langle \boldsymbol{\beta}_j^*(, \boldsymbol{z}^i), \boldsymbol{x}_{-j}^i \rangle, x_j^i \right) \right]$$

$$\leq \frac{1}{np} \sum_{i \in [n]} \left[ \sum_{j=1}^p \ell\left( \langle \boldsymbol{\beta}_i^{\text{opt}}(\boldsymbol{z}^i), \boldsymbol{x}_{-j}^i \rangle, x_j^i \right) - \sum_{j=1}^p \ell\left( \langle \boldsymbol{\beta}_j^*(\boldsymbol{z}^i), \boldsymbol{x}_{-j}^i \rangle, x_j^i \right) \right] \tag{15}$$

$$\leq \frac{L}{np} \sum_{i \in [n]} \left[ \sum_{j=1}^p \|\boldsymbol{\beta}_i^{\text{opt}} - \boldsymbol{\beta}_j^*\|_\infty \right] \tag{16}$$

$$\leq \underbrace{L \cdot \mathcal{E}_{\text{approx}}(\mathcal{F})}_{\text{approximation error}}.$$

Equation (15) is due to the fact that $\widehat{\boldsymbol{\beta}}$ is the empirical risk minimizer within $\mathcal{F}_{p \times (p-1)}$ and thus $\mathcal{R}_n(\widehat{\boldsymbol{\beta}}) \leq \mathcal{R}_n(\boldsymbol{\beta}^{\text{opt}})$. Finally, Equation (16) is derived by the Lipshitzness of $\ell(\cdot, \cdot)$. $\qquad \square$

### A.4  Proof of Corollary 1

*Proof.* Let $\varepsilon := \frac{p^2 L^2 d_p(\mathcal{F})}{\alpha n} \log(Lpn) \log\left(\frac{1}{\delta}\right)$, by (6) we have $\mathcal{R}(\widehat{\boldsymbol{\beta}}) - \mathcal{R}(\boldsymbol{\beta}^*) \lesssim L \cdot \mathcal{E}_{\text{approx}}(\mathcal{F}) + \varepsilon$, which could be expressed as:

$$\mathbb{E}_{\boldsymbol{x}, \boldsymbol{z}} \left[ \sum_{j=1}^p \ell\left( \langle \widehat{\boldsymbol{\beta}}_j(\boldsymbol{z}), \boldsymbol{x}_{-j} \rangle, x_j \right) - \sum_{j=1}^p \ell\left( \langle \boldsymbol{\beta}_j^*(\boldsymbol{z}), \boldsymbol{x}_{-j} \rangle, x_j \right) \right] \lesssim L \cdot \mathcal{E}_{\text{approx}}(\mathcal{F}) + \varepsilon.$$

By the $\alpha$-strongly convexity assumption (Assumption 1), namely,

$$\ell(a_1; b) - \ell(a_2; b) \geq \ell'(a; b)_{|a=a_2}(a_1 - a_2) + \frac{\alpha}{2}(a_1 - a_2)^2,$$

by setting $a_1 = \langle \widehat{\boldsymbol{\beta}}_j(\boldsymbol{z}), \boldsymbol{x}_{-j} \rangle$, $a_2 = \langle \boldsymbol{\beta}_j^*(\boldsymbol{z}), \boldsymbol{x}_{-j} \rangle$, $b = x_j$ and the optimality of $\boldsymbol{\beta}_j^*$, we have:

$$
\varepsilon + L \cdot \mathcal{E}_{\text{approx}} \geq \mathbb{E}_{\boldsymbol{x}, \boldsymbol{z}} \left[ \sum_{j=1}^p \ell \left( \langle \widehat{\boldsymbol{\beta}}_j(\boldsymbol{z}), \boldsymbol{x}_{-j} \rangle, x_j \right) - \sum_{j=1}^p \ell \left( \langle \boldsymbol{\beta}_j^*(\boldsymbol{z}), \boldsymbol{x}_{-j} \rangle, x_j \right) \right]
$$

$$
\geq \underbrace{\mathbb{E}_{\boldsymbol{x}, \boldsymbol{z}} \left[ \sum_{j=1}^p \left\{ \ell'(\langle \boldsymbol{\beta}_j, \boldsymbol{x}_{-j} \rangle, x_j)_{|\boldsymbol{\beta}_j = \boldsymbol{\beta}_j^*} \cdot \langle \widehat{\boldsymbol{\beta}}_j(\boldsymbol{z}) - \boldsymbol{\beta}_j^*(\boldsymbol{z}), \boldsymbol{x}_{-j} \rangle \right\} \right]}_{=0 \text{ by optimality of } \boldsymbol{\beta}_j^*}
$$

$$
+ \frac{\alpha}{2} \mathbb{E}_{\boldsymbol{x}, \boldsymbol{z}} \left[ \sum_{j=1}^p |\langle \widehat{\boldsymbol{\beta}}_j(\boldsymbol{z}) - \boldsymbol{\beta}_j^*(\boldsymbol{z}), \boldsymbol{x}_{-j} \rangle|^2 \right],
$$

which implies that :

$$
\mathbb{E}_{\boldsymbol{x}, \boldsymbol{z}} \left[ \sum_{j=1}^p |\langle \widehat{\boldsymbol{\beta}}_j(\boldsymbol{z}) - \boldsymbol{\beta}_j^*(\boldsymbol{z}), \boldsymbol{x}_{-j} \rangle|^2 \right] = \mathbb{E}_{\boldsymbol{z}} \left[ \sum_{j=1}^p \langle \widehat{\boldsymbol{\beta}}_j(\boldsymbol{z}) - \boldsymbol{\beta}_j^*(\boldsymbol{z}), \mathbb{E}_{\boldsymbol{x}} \left[ \boldsymbol{x}_{-j} \boldsymbol{x}_{-j}^\top \Big| \boldsymbol{z} \right] (\widehat{\boldsymbol{\beta}}_j(\boldsymbol{z}) - \boldsymbol{\beta}_j^*(\boldsymbol{z})) \rangle \right]
$$

$$
\leq \frac{2\varepsilon}{\alpha} + \frac{2L \cdot \mathcal{E}_{\text{approx}}}{\alpha}.
$$

$\square$

## A.5 Proof of Theorem 2

The result aims to balance the function approximation error $(O(\mathcal{E}_{\text{approx}}))$ and the generalization error $(O(\frac{d_P}{n}))$.

*Proof.* The major tool that we leverage is from Kohler & Langer (2021). In particular, if we invoke Theorem 2 from Kohler & Langer (2021) with $a = 1$, $p \simeq m$, $M \simeq \xi^{-1/2m}$, $L \simeq \xi^{-q/2m}$, $r \simeq (2e)^q \binom{m+q}{q} q^2$, we have that $\forall i \in [p], j \in [p]/i, \exists \beta_{jk}^{\text{opt}} \in \mathcal{F}, \|\beta_{jk}^{\text{opt}} - \beta_{jk}^*\|_\infty \leq \xi$ thus $\mathcal{E}_{\text{approx}}(\mathcal{F}) \leq \xi$. On the other hand, by Theorem 6 from Bartlett et al. (2019), we can bound the VC-dimension in terms of number of layers and neurons. In the case of a deep neural network (Theorem 2b from Kohler & Langer (2021)):

$$
d_P \leq \text{const} \cdot H^2 r^2 log(Hr)
$$

$$
\leq \text{const} \cdot \left( (2eM)^q \binom{m+q}{q} m \cdot q^2 \right)^2 \log(Hr)
$$

$$
\leq \text{const} \cdot (2e^{2+m/q}M)^{2q}(m^2 d^5)(\log(2e^{2+m/q}M) + \log(m) + \log(q)).
$$

It suffices to pick $M = \text{const} \cdot \xi^{-1/2m}$ and one can get that

$$
d_P \leq \text{const} \cdot -\xi^{-q/m} m^4 q^5 \log(\xi).
$$

Plugging the above into (6), one has:

$$
\mathcal{R}(\widehat{\boldsymbol{\beta}}) - \mathcal{R}(\boldsymbol{\beta}^*) \lesssim L\xi + \frac{L^2 \xi^{-q/m} m^4 q^5}{\alpha n} p^2 \log(1/\xi) \log(nLp) \log\left(\frac{1}{\delta}\right). \tag{17}
$$

To minimize the right-hand side of the above inequality, it suffices to choose

$$
\xi = \left( \frac{L m^4 q^6 p^2 \log^2(nLp) \log(p) \log(1/\delta) \log(1/\alpha)}{\alpha n} \right)^{\frac{m}{m+q}}. \tag{18}
$$

Finally, Equation (8) could be derived using a similar strong convexity argument as in Corollary 1 to bridge the gap between excess risk and the quality of $\boldsymbol{\beta}$: $\mathbb{E}_{\boldsymbol{z}}[\|\boldsymbol{\beta}(\boldsymbol{z}) - \boldsymbol{\beta}^*(\boldsymbol{z})\|_{A(\boldsymbol{z})}]$.

$\square$

## A.6 Extension of Theorem 2 with An Additional Assumption on the Loss Function

In this section we provide an extension of Theorem 2 under the following additional assumption:

**Assumption 6.** *Assume the following holds in addition for the loss function $\ell(\cdot, \cdot)$, namely its gradient is also Lipschitz:*

$$\ell(a_1; b) - \ell(a_2; b) \le \ell'(a; b)_{|a=a_2}(a_1 - a_2) + \frac{\lambda}{2}(a_1 - a_2)^2.$$

This is equivalent to assuming that the second derivative of the loss function $\ell''(\cdot)$ is upper bounded. Next, we state the result formally.

**Theorem 3.** Let $\mathcal{F}$ correspond to a family of fully connected neural networks with ReLU activation functions, with number of layers $H \simeq \xi^{-q/2m}$ and number of neurons $r \simeq (2e)^q \binom{m+q}{q} q^2$. Then, under the assumptions of Theorem 1, together with Assumption 4 with $m \lesssim q$ and Assumption 6, by setting

$$\xi = \left( \frac{L m^4 d^6 p^2 \log^2(nLp) \log(p) \log(1/\delta) \log(1/\alpha)}{\lambda \alpha n} \right)^{\frac{m}{2m+q}}, \tag{19}$$

the following holds with probability at least $1 - \delta$:

$$\mathcal{R}(\widehat{\boldsymbol{\beta}}) - \mathcal{R}(\boldsymbol{\beta}^*) \lesssim \lambda \xi^2, \quad \mathbb{E}_{\boldsymbol{z}} \left[ \frac{1}{p} \sum_{j=1}^{p} \|\widehat{\boldsymbol{\beta}}_j(\boldsymbol{z}) - \boldsymbol{\beta}_j^*(\boldsymbol{z})\|_{A_j(\boldsymbol{z})}^2 \right] \lesssim \frac{\lambda \xi^2}{\alpha}. \tag{20}$$

*Proof.* Using a similar argument as in the proof of Theorem 1, starting from (15):

$$
\begin{aligned}
\mathcal{R}_n(\widehat{\boldsymbol{\beta}}) - \mathcal{R}_n(\boldsymbol{\beta}^*) &= \frac{1}{np} \sum_{i \in [n]} \left[ \sum_{j=1}^{p} \ell\left( \langle \widehat{\boldsymbol{\beta}}_j(\boldsymbol{z}^i), \boldsymbol{x}_{-j}^i \rangle, x_j^i \right) - \sum_{j=1}^{p} \ell\left( \langle \boldsymbol{\beta}_j^*(, \boldsymbol{z}^i), \boldsymbol{x}_{-j}^i \rangle, x_j^i \right) \right] \\
&\le \frac{1}{np} \sum_{i \in [n]} \left[ \sum_{j=1}^{p} \ell\left( \langle \beta_i^{\mathrm{opt}}(\boldsymbol{z}^i), \boldsymbol{x}_{-j}^i \rangle, x_j^i \right) - \sum_{j=1}^{p} \ell\left( \langle \boldsymbol{\beta}_j^*(\boldsymbol{z}^i), \boldsymbol{x}_{-j}^i \rangle, x_j^i \right) \right] \\
&= \mathcal{R}_n(\boldsymbol{\beta}^{\mathrm{opt}}) - \mathcal{R}_n(\boldsymbol{\beta}^*).
\end{aligned}
$$

With inequality (13), the following holds with probability at least $1 - \delta$:

$$\mathcal{R}_n(\boldsymbol{\beta}^{\mathrm{opt}}) - \mathcal{R}_n(\boldsymbol{\beta}^*) \lesssim \mathcal{R}(\boldsymbol{\beta}^{\mathrm{opt}}) - \mathcal{R}(\boldsymbol{\beta}^*) + \frac{d_P(\mathcal{F})}{\alpha n} L^2 p^2 \log(nLp) \log\left( \frac{1}{\delta} \right).$$

Together with (14), we have:

$$\mathcal{R}(\widehat{\boldsymbol{\beta}}) - \mathcal{R}(\boldsymbol{\beta}^*) \lesssim \mathcal{R}(\boldsymbol{\beta}^{\mathrm{opt}}) - \mathcal{R}(\boldsymbol{\beta}^*) + \frac{d_P(\mathcal{F})}{\alpha n} L^2 p^2 \log(nLp) \log\left( \frac{1}{\delta} \right). \tag{21}$$

For the term $\mathcal{R}(\boldsymbol{\beta}^{\mathrm{opt}}) - \mathcal{R}(\boldsymbol{\beta}^*)$ on the RHS, it satisfies

$$
\begin{aligned}
\mathcal{R}(\boldsymbol{\beta}^{\mathrm{opt}}) - \mathcal{R}(\boldsymbol{\beta}^*) &= \frac{1}{np} \mathbb{E}_{\boldsymbol{x}, \boldsymbol{z}} \left[ \sum_{j=1}^{p} \ell\left( \langle \boldsymbol{\beta}_j^{\mathrm{opt}}(\boldsymbol{z}), \boldsymbol{x}_{-j} \rangle, x_j \right) - \sum_{j=1}^{p} \ell\left( \langle \boldsymbol{\beta}_j^*(\boldsymbol{z}), \boldsymbol{x}_{-j} \rangle, x_j \right) \right] \\
&\le \frac{1}{np} \mathbb{E}_{\boldsymbol{x}, \boldsymbol{z}} \Bigg[ \underbrace{\sum_{j=1}^{p} \left\{ \ell'(\langle \boldsymbol{\beta}_j, \boldsymbol{x}_{-j} \rangle, x_j)_{|\boldsymbol{\beta}_j = \boldsymbol{\beta}_j^*} \cdot \langle \boldsymbol{\beta}_j^{\mathrm{opt}}(\boldsymbol{z}) - \boldsymbol{\beta}_j^*(\boldsymbol{z}), \boldsymbol{x}_{-j} \rangle \right\}}_{=0, \text{ by the optimality of } \boldsymbol{\beta}_j^*} \Bigg] \\
&\quad + \frac{\lambda}{2} \cdot \frac{1}{np} \mathbb{E}_{\boldsymbol{x}, \boldsymbol{z}} \left[ \sum_{j=1}^{p} |\langle \boldsymbol{\beta}_j^{\mathrm{opt}}(\boldsymbol{z}) - \boldsymbol{\beta}_j^*(\boldsymbol{z}), \boldsymbol{x}_{-j} \rangle|^2 \right] \\
&\le \frac{\lambda}{2np} \sum_{i \in [n]} \left[ \sum_{j=1}^{p} \|\boldsymbol{\beta}_i^{\mathrm{opt}} - \boldsymbol{\beta}_j^*\|_\infty^2 \right] \\
&\le \frac{\lambda}{2} \cdot \mathcal{E}_{\mathrm{approx}}(\mathcal{F})^2.
\end{aligned}
$$

Plugging the above inequality into (21) we have:

$$\mathcal{R}(\widehat{\boldsymbol{\beta}}) - \mathcal{R}(\boldsymbol{\beta}^*) \lesssim \frac{\lambda}{2} \cdot \mathcal{E}_{\mathrm{approx}}(\mathcal{F})^2 + \frac{d_P(\mathcal{F})}{\alpha n} L^2 p^2 \log(nLp) \log\left(\frac{1}{\delta}\right).$$

Let $\mathcal{E}_{\mathrm{approx}}(\mathcal{F}) := \xi$, by plugging inequality (17) into above inequality, we have:

$$\mathcal{R}(\widehat{\boldsymbol{\beta}}) - \mathcal{R}(\boldsymbol{\beta}^*) \lesssim \frac{\lambda}{2} \cdot \xi^2 + \frac{\xi^{-q/m} m^4 q^5}{\alpha n} L^2 p^2 \log(nLp) \log\left(\frac{1}{\delta}\right).$$

To minimize the RHS of above inequality, if suffices to choose

$$\xi \simeq \left( \frac{Lm^4 q^6 p^2 \log^2(nLp) \log(p) \log(1/\delta) \log(1/\alpha)}{\lambda \alpha n} \right)^{\frac{m}{2m+q}}.$$

$\square$

*Remark* 10. The choice of $\xi$ in (19) implies that the $\mathcal{R}(\widehat{\boldsymbol{\beta}}) - \mathcal{R}(\boldsymbol{\beta}^*)$ converges at a rate of the order $n^{-2m/(2m+q)}$ which matches the rate from Kohler & Langer (2021).

### A.7 Proof of Corollary 2

From inequality (7) we have:

$$\mathbb{E}_{\boldsymbol{z}}\left[ \sum_{j=1}^p \langle \widehat{\boldsymbol{\beta}}_j(\boldsymbol{z}) - \boldsymbol{\beta}_j^*(\boldsymbol{z}), \, A_j(\boldsymbol{z})(\widehat{\boldsymbol{\beta}}_j(\boldsymbol{z}) - \boldsymbol{\beta}_j^*(\boldsymbol{z})) \rangle \right]$$

$$\lesssim \frac{L^2 d_P(\mathcal{F})}{\alpha^2 (\underline{\beta} - \bar{\beta})^2 n} p^2 \log(nLp) \log\left(\frac{1}{\delta}\right) + \frac{L \cdot \mathcal{E}_{\mathrm{approx}}(\mathcal{F})}{\alpha(\underline{\beta} - \bar{\beta})^2}.$$

Using Assumption 5, we have $\langle \widehat{\boldsymbol{\beta}}_j(\boldsymbol{z}) - \boldsymbol{\beta}_j^*(\boldsymbol{z}), \, A_j(\boldsymbol{z})(\widehat{\boldsymbol{\beta}}_j(\boldsymbol{z}) - \boldsymbol{\beta}_j^*(\boldsymbol{z})) \rangle \geq \frac{1}{\gamma} \|\widehat{\boldsymbol{\beta}}_j(\boldsymbol{z}) - \boldsymbol{\beta}_j^*(\boldsymbol{z})\|_2^2$ for each $j = 1, \cdots, p$ and therefore:

$$\mathbb{E}_{\boldsymbol{z}}\left[ \sum_{j=1}^p \sum_{k \neq j} \left( \hat{\beta}_{jk}(\boldsymbol{z}) - \beta_{jk}^*(\boldsymbol{z}) \right)^2 \right] \lesssim \frac{L^2 d_P(\mathcal{F})}{\alpha^2 \gamma (\underline{\beta} - \bar{\beta})^2 n} p^2 \log(nLp) \log\left(\frac{1}{\delta}\right) + \frac{L \cdot \mathcal{E}_{\mathrm{approx}}(\mathcal{F})}{\alpha \gamma (\underline{\beta} - \bar{\beta})^2}.$$

On the other hand, by the fact that the margin between strong and weak edges is bounded away from zero, namely, $(\underline{\beta} - \bar{\beta}) \geq \phi > 0$, we can select a threshold $\tau = \eta \bar{\beta} + (1 - \eta)\underline{\beta}$ and measure the edge-wise guarantee using the binary risk: $\mathbb{1}\{|\widehat{\beta}_{jk}(\boldsymbol{z})| \geq \tau\} \neq \mathbb{1}\{|\beta_{jk}^*(\boldsymbol{z})| \geq \underline{\beta}\}$.

Next, we establish the following inequality:

$$\min\{\eta^2, (1-\eta)^2\}(\underline{\beta} - \bar{\beta})^2 \left\{ \mathbb{1}\{|\widehat{\beta}_{jk}(\boldsymbol{z})| \geq \tau\} \neq \mathbb{1}\{|\beta_{jk}^*(\boldsymbol{z})| \geq \underline{\beta}\} \right\} \leq (\widehat{\beta}_{jk}(\boldsymbol{z}) - \beta_{jk}^*(\boldsymbol{z}))^2. \tag{22}$$

The starting point is the following decomposition:

$$(\widehat{\beta}_{jk}(\boldsymbol{z}) - \beta_{jk}^*(\boldsymbol{z}))^2 = \underbrace{\left\{ \mathbb{1}\{|\widehat{\beta}_{jk}(\boldsymbol{z})| \geq \tau\} \neq \mathbb{1}\{|\beta_{jk}^*(\boldsymbol{z})| \geq \underline{\beta}\} \right\} (\widehat{\beta}_{jk}(\boldsymbol{z}) - \beta_{jk}^*(\boldsymbol{z}))^2}_{\text{Term I}}$$

$$+ \underbrace{\left\{ \mathbb{1}\{|\widehat{\beta}_{jk}(\boldsymbol{z})| \geq \tau\} = \mathbb{1}\{|\beta_{jk}^*(\boldsymbol{z})| \geq \underline{\beta}\} \right\} (\widehat{\beta}_{jk}(\boldsymbol{z}) - \beta_{jk}^*(\boldsymbol{z}))^2}_{\text{Term II}}.$$

To show inequality (22), it suffices to bound Term I. In case $\mathbb{1}\{|\widehat{\beta}_{jk}(\boldsymbol{z})| \geq \tau\} \neq \mathbb{1}\{|\beta_{jk}^*(\boldsymbol{z})| \geq \underline{\beta}\}$, we have either $|\widehat{\beta}_{jk}(\boldsymbol{z})| \geq \tau, |\beta_{jk}^*(\boldsymbol{z})| \leq \underline{\beta}$, or $|\widehat{\beta}_{jk}(\boldsymbol{z})| \leq \tau, |\beta_{jk}^*(\boldsymbol{z})| \geq \underline{\beta}$. In both cases, we have $\min\{\eta^2, (1-\eta)^2\}(\underline{\beta} - \bar{\beta})^2 \leq (\widehat{\beta}_{jk}(\boldsymbol{z}) - \beta_{jk}^*(\boldsymbol{z}))^2$, and hence

$$\min\{\eta^2, (1-\eta)^2\}(\underline{\beta} - \bar{\beta})^2 \left\{ \mathbb{1}\{|\widehat{\beta}_{jk}(\boldsymbol{z})| \geq \tau\} \neq \mathbb{1}\{|\beta_{jk}^*(\boldsymbol{z})| \geq \underline{\beta}\} \right\} \leq (\widehat{\beta}_{jk}(\boldsymbol{z}) - \beta_{jk}^*(\boldsymbol{z}))^2.$$

The last expression implies that

$$\min\{\eta^2, (1-\eta)^2\}(\underline{\beta} - \bar{\beta})^2 \left\{ \sum_{j=1}^{p} \sum_{k \neq j} \mathbb{1}\{|\widehat{\beta}_{jk}(\boldsymbol{z})| \geq \tau\} \neq \mathbb{1}\{|\beta_{jk}^*(\boldsymbol{z})| \geq \underline{\beta}\} \right\} \leq \sum_{j=1}^{p} \sum_{k \neq j} (\hat{\beta}_{jk}(\boldsymbol{z}) - \beta_{jk}^*(\boldsymbol{z}))^2.$$

Consequently, it follows that

$$\mathbb{E}_{\boldsymbol{z}} \left[ \sum_{j=1}^{p} \sum_{k \neq j} \mathbb{1}\{|\widehat{\beta}_{jk}(\boldsymbol{z})| \geq \tau\} \neq \mathbb{1}\{|\beta_{jk}^*(\boldsymbol{z})| \geq \underline{\beta}\} \right]$$

$$\lesssim \frac{L^2 d_P(\mathcal{F})}{\alpha^2 \min\{\eta^2, (1-\eta)^2\}\gamma(\underline{\beta} - \bar{\beta})^2 n} p^2 \log(nLp) \log\left(\frac{1}{\delta}\right) + \frac{L \cdot \mathcal{E}_{\mathrm{approx}}(\mathcal{F})}{\alpha\gamma \min\{\eta^2, (1-\eta)^2\}(\underline{\beta} - \bar{\beta})^2}.$$

## B  Additional Details and Results for Synthetic Data Experiments

**Additional illustration for the data generating process.**  Figure 2 provides visualization for candidate skeletons $\Psi_l$'s that are used in settings G1, G2, N1 and N2, and selected $\Theta^i$'s that are obtained as the convex combination of the candidates, depending on the value of their corresponding $\boldsymbol{z}^i$'s.

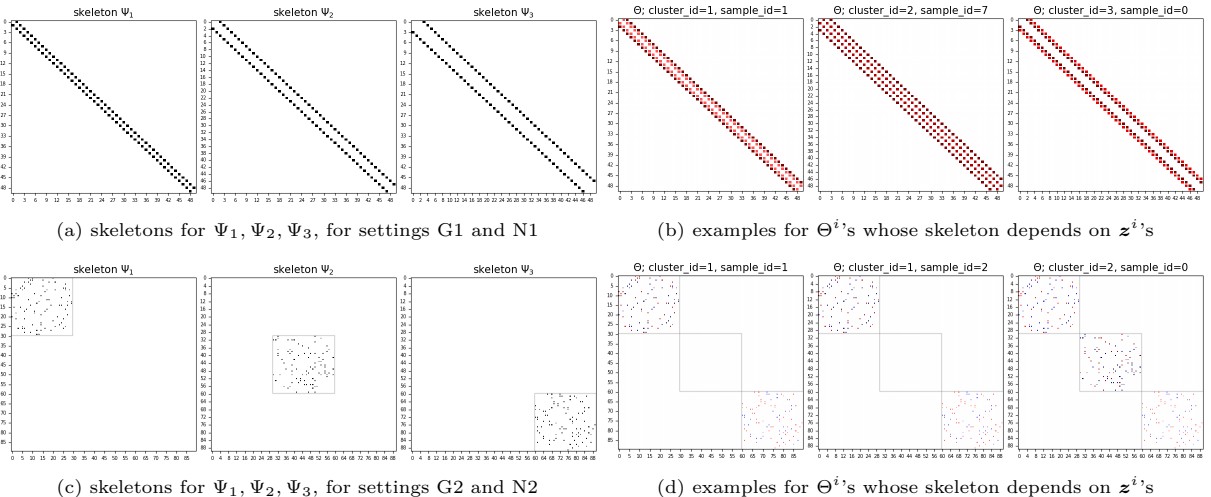

(a) skeletons for $\Psi_1, \Psi_2, \Psi_3$, for settings G1 and N1      (b) examples for $\Theta^i$'s whose skeleton depends on $\boldsymbol{z}^i$'s

(c) skeletons for $\Psi_1, \Psi_2, \Psi_3$, for settings G2 and N2      (d) examples for $\Theta^i$'s whose skeleton depends on $\boldsymbol{z}^i$'s

Figure 2: Pictorial illustration for settings G1,G2,N1,N2. Left: candidate skeletons $\Psi_1, \Psi_2, \Psi_3$. Right: different $\Theta^i$'s that are obtained from candidate skeletons, with the exact mixing depending on the values of the corresponding $\boldsymbol{z}^i$'s; their diagonals are suppressed for visualization purpose.

Figure 3a provides visualizations for the candidate skeletons $B_1, B_2$ whose corresponding graphs have a tree structure[15], and their convex combination (no longer corresponds to a tree); they serve as the skeletons for the DAGs. Figure 3b shows the resulting $\Theta^i$'s after moralization. The color (red/blue) and shade respectively reflect the sign (positive/negative) and the magnitude of the entries in the moralized graphs for the linear case where the exact values can be calculated.

Of particular note, in the special case where the DAG possesses a tree structure, the moralized graph is equivalent to removing the direction of the edges in the DAG, and therefore the skeleton matrix can be obtained by adding the "transposed" entries to that of the DAG; e.g., see the first two plots in Figure 3b. Otherwise, in addition to the "transposed" entries, there are additional "married" entries as a result of connecting nodes having common children. As it can be seen from the rightmost plot, these "married" entries (the very faint ones in the middle of the heatmap) tend to be extremely weak in magnitude, and pose challenges for estimating them due to the very low signal-to-noise. Finally, for the non-linear case, one can

---

[15]Note that by definition of the DAG, the nodes possess a topological ordering and thus the corresponding matrix can be written as a triangular one after reordering; here we are showing the skeleton matrix after such reordering.

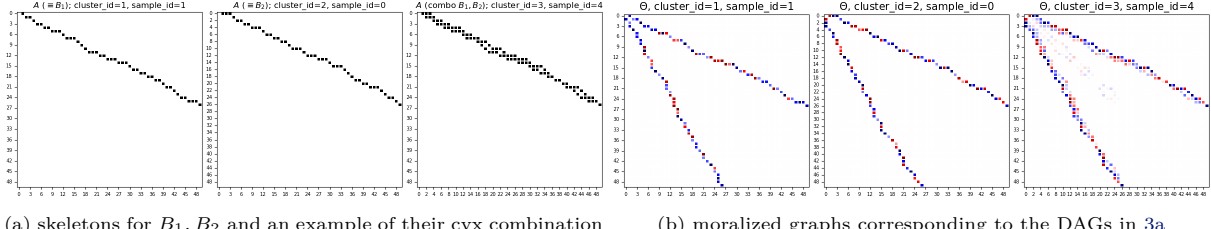

(a) skeletons for $B_1, B_2$ and an example of their cvx combination   (b) moralized graphs corresponding to the DAGs in 3a

Figure 3: Pictorial illustration for setting D1. Left: candidate skeletons $B_1, B_2$ and an example of their convex combination. Right: Moralized graphs for the respective DAGs, with their diagonals suppressed for visualization purpose.

obtain values of the moralized graph *approximately*, by considering linear approximation to the $f_{jk}^i$'s (recall that $x_j^i = \sum_{k \in \text{pa}(j)} f_{jk}^i(x_k^i) + \epsilon_j$), and they exhibit qualitatively similar patterns to the linear case.

**Additional simulation results.** Table 4 presents additional metrics for the DNN-based covariate-dependent graphical model estimation method. In particular, given the sparse nature of the underlying true graphs, we report the F1 score ($2 * \frac{\text{precision} * \text{recall}}{\text{precision} + \text{recall}}$) and balanced accuracy ($\frac{\text{sensitivity} + \text{specificity}}{2}$) when the estimated graphs are thresholded at different levels. The result shows that for practical purposes, practitioners can effectively recover a sparse skeleton that is close to the truth by applying a reasonably small thresholding.

Table 4: F1 score (F1) and Balance Accuracy (BA) for the estimated graphs by **DNN-GCM** at different thresholding levels.

|           | G1  |     | G2  |     | N1  |     | N2  |     | D1  |     | D2  |     |
|-----------|-----|-----|-----|-----|-----|-----|-----|-----|-----|-----|-----|-----|
| threshold | F1  | BA  | F1  | BA  | F1  | BA  | F1  | BA  | F1  | BA  | F1  | BA  |
| 0.010     | 0.23| 0.71| 0.15| 0.84| 0.23| 0.71| 0.15| 0.84| 0.23| 0.76| 0.24| 0.76|
| 0.025     | 0.49| 0.91| **0.72**| **0.98**| 0.50| 0.91| **0.71**| **0.96**| 0.56| 0.91| **0.63**| **0.85**|
| 0.050     | 0.83| 0.98| 0.70| 0.81| 0.89| 0.99| 0.65| 0.77| **0.75**| **0.89**| 0.56| 0.73|
| 0.075     | 0.91| 0.98| 0.57| 0.71| 0.95| 0.98| 0.56| 0.69| 0.70| 0.84| 0.32| 0.60|
| 0.100     | **0.94**| **0.97**| 0.48| 0.66| **0.93**| **0.96**| 0.41| 0.63| 0.65| 0.79| 0.02| 0.51|

Note: reported metrics are first averaged across test samples for a single experiment, then averaged across experiments on 5 data replicates.

Table 5 presents additional results for settings D1 and D2, where instead of evaluating the performance against the true moralized graph $\Theta^i$, we evaluate against a "pseudo" moralized graph $\tilde{\Theta}^i$ where we treat edges that are present due to married nodes as if they were non-existent. As previously mentioned, these edges typically exhibit very small magnitude and hence are difficult to recover. We expect the performance to improve slightly if the comparison were done against the $\tilde{\Theta}^i$'s, which is indeed the case: AUROC improves by around 3% and AUPRC improves by around 5-7% (see, e.g., the last two rows in Table 3 for a comparison).

Table 5: Performance evaluation for DAG-based settings against the pseudo moralized graph without edges from married nodes.

|    | **DNN CGM** |         | RegGMM  |         | glasso - est. by cluster | | nodewise Lasso - est. by cluster | |
|----|-------------|---------|---------|---------|--------------|--------------|--------------|--------------|
|    | AUROC       | AUPRC   | AUROC   | AUPRC   | AUROC        | AUPRC        | AUROC        | AUPRC        |
| D1 | 0.98 (0.003)| 0.81 (0.028)| 0.97 (0.006)| 0.81 (0.028)| 0.97 (0.003)| 0.50 (0.007)| 0.99 (0.002)| 0.89 (0.020)|
| D2 | 0.96 (0.013)| 0.74 (0.032)| 0.91 (0.015)| 0.65 (0.023)| 0.94 (0.009)| 0.44 (0.002)| 0.92 (0.002)| 0.52 (0.110)|

*Remark* 11. For DNN-based method, the performance can improve when training is done with a larger sample size. In particular, for challenging settings such as D1 and D2, for estimates obtained from a model trained using 30,000 samples, when they are evaluated against the true graph $\Theta^i$, AUROC is given by 0.97 (0.004) and 0.95 (0.007) respectively for D1 and D2, and AUPRC is given by 0.91 (0.004) and 0.75 (0.015). When the estimates are evaluated against the pseudo moral graph $\tilde{\Theta}^i$ (so that extremely weak entries from married nodes are not counted toward the skeleton), AUROC reaches around 0.99 for both settings and AUPRC reaches 0.95 and 0.82, resp.

Finally, Table 6 shows the metrics for glasso and nodewise Lasso when they perform estimation on the full set of samples; note that in the absence of covariate-dependent estimation methods, this would be how the two methods perform in practice since the partitioning would not be known apriori.

Table 6: Evaluation for glasso and nodewise Lasso on full samples without partitioned by clusters.

| | | G1 | G2 | N1 | N2 | D1 | D2 | D1 - pseudo moralized graph | D2 - pseudo moralized graph |
|---|---|---|---|---|---|---|---|---|---|
| glasso | AUROC | 0.95 | 0.98 | 0.96 | 0.98 | 0.91 | 0.89 | 0.95 | 0.94 |
| | AUPRC | 0.47 | 0.46 | 0.47 | 0.45 | 0.39 | 0.37 | 0.43 | 0.41 |
| nodewise Lasso | AUROC | 0.97 | 0.98 | 0.97 | 0.97 | 0.92 | 0.88 | 0.97 | 0.94 |
| | AUPRC | 0.66 | 0.67 | 0.66 | 0.47 | 0.55 | 0.43 | 0.62 | 0.53 |

Note: values correspond to the average over experiments on 5 data replicates. The last 2 columns show the evaluation against pseudo moralized graphs under DAG settings.

Compared with the results in Table 3, nodewise Lasso exhibits material improvement when the dependency on the covariate is linear/mildly non-linear, whereas glasso seems more susceptible to the mis-specification, induced by the varying magnitude across samples, as manifested by a much smaller improvement in performance in the case where estimation is conducted on partitioned samples.

## C   Additional Real Data Experiments

In this section, we present results obtained from applying the DNN-based covariate-dependent graphical model to a finance dataset involving S&P 100 stocks. Such a model estimates the *partial correlation* across stocks, while conditioning on covariates that correspond to the broad market condition.

**The S&P 100 constituent dataset.**   We examine the inter-connectedness of the S&P 100 Index[16] constituent stocks under different market conditions. These constituent stocks correspond to 100 major blue chip companies in the United States and span various industry sectors. In the sequel, stocks and tickers may be used interchangeably, and they both correspond to the nodes of the network of interest.

**Data collection and preprocessing.**   We collect daily stock return (calculated based on adjusted close price) data for those that are components of the S&P 100 Index as of 2023-12-29. At the initial data gathering stage, the first historical date is set to 2000-01-02, and the last to 2023-12-29. We require all tickers to have valid data on the first historical date, and the set of tickers are further filtered to ensure this; tickers such as `ABBV`, `GM`, `GOOG`, `META` etc are therefore excluded from subsequent analyses. After this filtering step, the remaining set encompasses 79 tickers and thus the size of the network $p = 79$.

To obtain samples for $\boldsymbol{x}$, we consider beta-adjusted SPX residual returns, so that the market information is profiled out from the mean structure. Concretely, as a data-preprocessing step, let $m^i$ be the S&P 500 Index return on day $i$, which is a proxy for the market return of US large-cap companies; for each ticker indexed by $j$, let $\check{x}_j^i$ be its raw return. We first estimate its beta $\beta_j^i$ by regressing its return against that of the S&P 500 Index, over a lookback window of 252 days; the beta-adjusted residual return is then given by $x_j^i := \check{x}_k^i - \widehat{\beta}_j^i m^i$. This gives rise to $\{x_j^i, j = 1, \cdots, p\}$ for all tickers. After this pre-processing step, the residual return data (namely, the $\boldsymbol{x}^i$'s) effectively starts from 2001-01-02 and there are a total number of 5785 observations. Note that the processed data exhibit de minimis serial correlation and the $\boldsymbol{x}^i$'s can be regarded as i.i.d. for practical purposes. For $\boldsymbol{z}$, we simply consider a set of variables that can reflect the high-level market condition of the day, and it encompasses the returns of the S&P 500 Index, the Nasdaq Composite Index, respectively, and VIX. As such, $q = 3$. Data is further split into train/val/test periods that respectively span 2001-2017, 2018-2019, 2020 onwards. Note that such a split is more for the purpose of demonstrating that the trained model is capable of producing interpretable results when applied to unseen data (e.g., COVID period); however, results are presented for all periods to demonstrate the patterns the proposed method uncover.

**Results.**   Figure 4 shows various graph connectivity metrics over time (left) and estimated graphs on selected dates (right), after thresholding. The observation is 3-fold: (1) the estimated networks exhibit

---

[16]https://www.spglobal.com/spdji/en/indices/equity/sp-100/#overview

significantly higher connectivity during volatile market conditions[17]. (2) Under normal market conditions, intra-sector connectivity level is higher, as manifested by the corresponding heatmap showing a block-diagonal pattern; however, in adverse conditions, such a pattern is diluted due to increased connectivity across the board. (3) Overall, the graph dynamics are slow-moving in the sense that the estimated graphs usually show high concordance for dates that are not far apart.

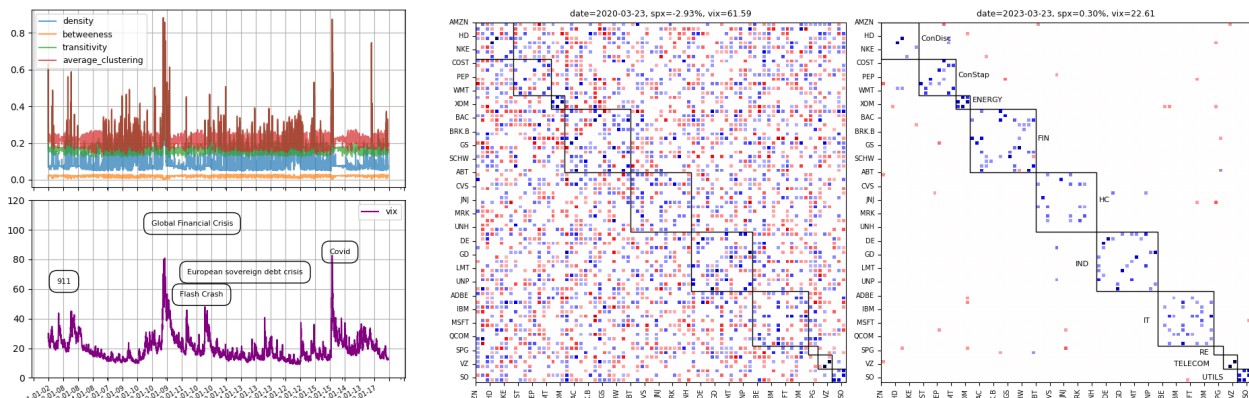

(a) graph connectivity metrics (top) and VIX level (bottom)

(b) heatmaps for estimated graphs on selected high-vix (left, 2020-03-23) and low-vix (right, 2023-03-23) days.

Figure 4: **Left panel**: various connectivity metrics of the estimated graphs (top) and the corresponding VIX level (bottom), from the beginning of 2001 to the end of 2023. Notably events that significantly increased market volatility have been marked. **Right panel**: heatmaps of the estimated graphs on two representative dates that respectively have high and low-VIX, with red cells indicating positive partial correlation and blue cells negative ones.

Figure 5 presents the *average* partial correlation graph of high and normal VIX days in the test period, respectively, where high VIX days encompass dates on which VIX exceeded the 90% percentile of the period in question and normal VIX ones correspond to those that were below median.

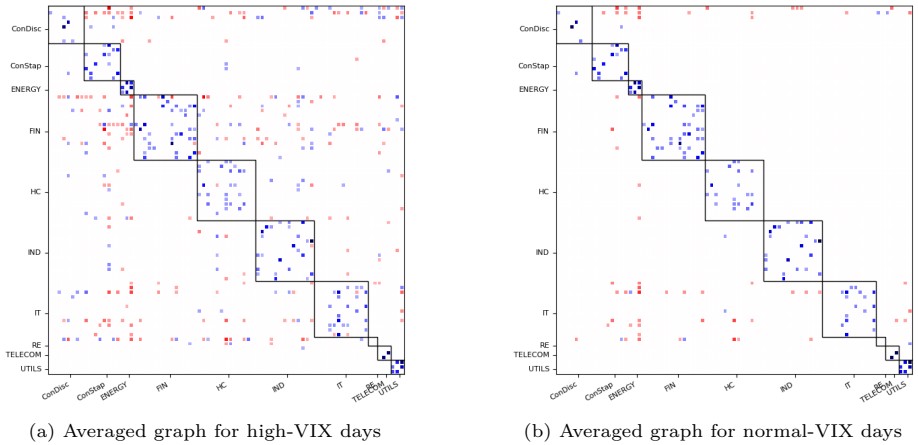

(a) Averaged graph for high-VIX days

(b) Averaged graph for normal-VIX days

Figure 5: Averaged partial correlation graphs for high-VIX (left) and normal-VIX (right) days, with red cells indicating positive partial correlation and blue cells negative ones.

After the averaging step, the increased connectivity during the high-VIX period is not as pronounced as that on the extremes (e.g., 2020-03-23) where the intra-sector pattern no longer stands out. Nonetheless, compared with the average graph from the normal-VIX days, the connectivity is still at an elevated level, as manifested by the additional inter-sector connections while the intra-sector ones stay comparable.

---

[17]Similar patterns have been reported in existing financial econometrics literature (e.g., Billio et al., 2012; Diebold & Yılmaz, 2014), albeit models/methods therein examine different quantities rather than the precision matrix. Note also that these methods do not support sample-specific estimates and thus their results were obtained through rolling window analyses.

Finally, note that we refrain from dictating any use case of the estimated graphs, which highly depends on the specific context and application.

## D  Graphical Model Preliminaries

### D.1  Decomposition of a Gaussian Multivariate Distribution into Clique-wise Potential Functions According to a Graph $G$.

Consider a Gaussian random vector $\boldsymbol{x} \in \mathbb{R}^p$ with mean vector $\mu = 0$ and covariance matrix $\Sigma$ assumed to be positive definite. Its joint distribution function is given by

$$\mathbb{P}(\boldsymbol{x}) = \frac{1}{\sqrt{(2\pi)^p \det(\Sigma)}} \exp\left(-\frac{1}{2}\boldsymbol{x}^\top \Sigma^{-1} \boldsymbol{x}\right).$$

Then,

$$\log(\boldsymbol{x}) = C + \sum_{j,k=1}^{p} x_j x_k \Sigma_{jk}^{-1},$$

and hence the potential functions over *pairwise cliques* correspond to

$$\varphi_{(j,k)}(\boldsymbol{x}) = x_j x_k \Sigma_{jk}^{-1};$$

they are functions of $\Sigma$ which is the parameter of the specific Gaussian distribution under consideration. Further, it can be easily seen that $x_j \perp\!\!\!\perp x_k \,|\, \boldsymbol{x}_{-\{j,k\}}$, if and only if $\Sigma_{jk}^{-1} = 0$. Hence, to estimate a Gaussian graphical model from data, it suffices to estimate the inverse covariance matrix and all the pairwise conditional independent relationships encoded by a graph $G$ correspond to the zero elements of $\Sigma^{-1}$.

It has be further shown that the Gaussian graphical model can also be estimated based on regression techniques; namely, by estimating the regression coefficients of the following model (Meinshausen & Bühlmann, 2006), a procedure referred to as *neighborhood selection*:

$$x_j = \sum_{k=1, k \neq j}^{p} \beta_{jk} x_k + \varepsilon_j,$$

where $\beta_{jk} := -\Sigma_{jk}^{-1}/\Sigma_{jj}^{-1}$.

### D.2  An Exponential Family based Node-conditional Graphical Model

Consider a random vector $\boldsymbol{x} \in \mathbb{R}^p$ whose joint distribution is $\mathbb{P}(\boldsymbol{x})$. Further, consider the following collection of node conditional distributions from the *univariate exponential family*, where the neighbors of a node $j$ according to an underlying graph $G$ are denoted by $\mathcal{N}_G(j)$:

$$\mathbb{P}(x_j | \boldsymbol{x}_{-j}) \propto \exp\left\{g(x_j)\left(\theta_j + \sum_{k \in \mathcal{N}_G(j)} \theta_{jk} g(x_k) + \sum_{k,\ell \in \mathcal{N}_G(j)} \theta_{jk\ell} g(x_k) g(x_l)\right.\right. \tag{23}$$

$$\left.\left. + \cdots + \sum_{m_1,\cdots,m_C \in \mathcal{N}_G(j)} \theta_{jm_1\cdots,m_c} \prod_{c=1}^{C} g(x_c))\right)\right\},$$

where the multivariate *canonical parameter* $\boldsymbol{\theta}$ is defined as linear combinations of up to $C$-order products of positive univariate functions $g(x_k)$ of neighboring nodes to $j$ according to the graph $G$.

An application of the Hammerseley-Clifford theorem (see Section D.3) and some algebra (Yang et al., 2015) show that the collection of the node conditional distributions in (23) define a proper graphical model with respect to $G$.

### D.3 Markov Properties of Graphical Models

The conditional independence/dependence relationships defined by a graphical model are a consequence of the Markov properties encoded in the graph $G$. Specifically, the probability distribution $\mathbb{P}$ satisfies the *pairwise Markov property* with respect to $G$, if $x_j \perp\!\!\!\perp x_k \mid \boldsymbol{x}_{-(j,k)}$ for all $\{k, j\} \notin E$ i.e., variables $x_j$ and $x_k$ are *conditional independent* given all the other variables, if they are *not* connected according to the edge set $E$ of $G$. It satisfies the *local Markov property* with respect to $G$, if $x_k \perp\!\!\!\perp \boldsymbol{x}_{V-(\mathrm{ne}_G(j)\cup\{k\})} \mid \boldsymbol{x}_{\mathrm{ne}_G(j)}$; i.e., the conditional distribution of variable $x_j$ given all its neighbors is independent of any other nodes in $G$. Finally, $\mathbb{P}$ satisfies the *global Markov property* with respect to $G$, if for any subsets $A, B, C$ of $V$ such that $C$ *separates* $A$ and $B$ (i.e., every path between a node in $A$ and a node in $B$ contains a node in $C$), the following relationship holds: $\boldsymbol{x}_A \perp\!\!\!\perp \boldsymbol{x}_B \mid \boldsymbol{x}_C$. It can be shown that the global Markov property implies the local one and both imply the pairwise one.

Next, we elaborate on the relationship between the joint probability distribution $\mathbb{P}(\boldsymbol{x})$ of the random vector $\boldsymbol{x}$ and the underlying graph $G$ regarding conditional independence/dependence relationships. The connection between the graph $G$ and the probability distribution $\mathbb{P}(\mathbf{x})$ comes through the concept of graph factorization. Specifically, let $\mathcal{C}(G)$ denote the set of all cliques of $G$. Then, $\mathbb{P}(\boldsymbol{x})$ *factorizes* with respect to $G$, if it can be written as $\mathbb{P}(\boldsymbol{x}) = \frac{1}{Z} \prod_{C \in \mathcal{C}(G)} \phi_C(\boldsymbol{x}_C)$, with $\phi_C > 0$ for all $C \in \mathcal{C}(G)$; i.e., the joint distribution of $\boldsymbol{x}$ can be written as a product of positive functions (called potential functions in the literature) that depend only on a subset of the random variables in the clique. However, the concept of graph factorization does not imply per se conditional independence/dependence relationships between subsets of the random vector $\mathbf{x}$. Such relationships are associated through the various Markov properties of $G$, as outlined above (see also Lauritzen, 1996, for an indepth presentation). Hence, if $\mathbb{P}(\boldsymbol{x})$ factorizes over $G$, and $G$ possesses a Markov property, then conditional independence relationships for subsets of $\boldsymbol{x}$ are present under $\mathbb{P}$. The reverse relationship, namely that if $\mathbb{P}$ satisfies a Markov property with respect to $G$, then it also factorizes with respect to $G$, is established by the famous Hammerseley-Clifford theorem, provided that $\mathbb{P}(\boldsymbol{x})$ possesses a positive and continuous density (Besag, 1974).

## E  Notes on Implementation

**Implementation of the neural network.**  For all synthetic and real data experiments, we use fully connected MLPs (i.e., stacked modules consisting of Linear-ReLU-Dropout layers) with residual connections as the underlying architecture to parameterize the $\beta_{jk}(\cdot)$'s. In particular, we adopt the implementation scheme as mentioned in Remark 1, where we use a single neural network that takes $\boldsymbol{z}$ as the input (i.e., input dim is $q$), but with a multi-dimensional output head that produces outputs for all $\beta_{jk}(\cdot), j, k \in [p], k \neq j$ (i.e., output dim is $p(p-1)$).

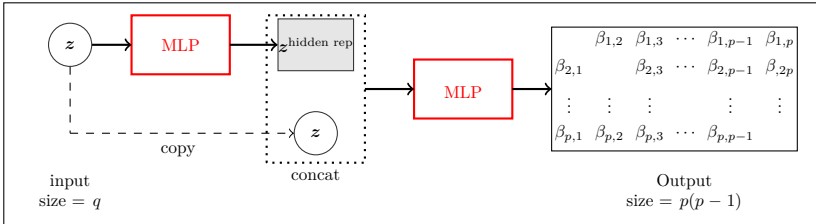

Figure 6: Diagram for the $\boldsymbol{\beta}(\boldsymbol{z}) : \mathbb{R}^q \mapsto \mathbb{R}^{p(p-1)}$ network; an MLP block consists of Linear-ReLU-Dropout layers.

Overall, we observe that the above-mentioned implementation scheme is fairly robust to the choice of hyper-parameters; in addition, it shows material improvement in performance when compared with a single MLP block without the residual connection, when $q$ becomes moderately large. Other "intermediate" sharing-backbone schemes have been experimented, yet the performance is generally inferior and more sensitive to the choice of hyper-parameters. During training, we fix batch size at 512 with gradient clipped at 1.0, while other hyperparameters are tuned over a hyper-grid, with the best set determined based the performance on

a validation set of 1000 samples, with MSE as the criterion. All experiments are done on a NVIDIA RTX A5000 GPU.

Table 7: hyper-parameters for the MLPs and model training

|  | hidden layer size/dropout | learning rate | scheduler type | scheduler stepsize(milestones) / decay | epochs |
|---|---|---|---|---|---|
| G1 | [128, 64], [128] / 0.3 | 0.0005 | StepLR | 20/0.25 | 50 |
| G2 | [128, 64], [128] / 0.3 | 0.0005 | StepLR | 20/0.25 | 80 |
| N1 | [128, 64], [128] / 0.3 | 0.0005 | StepLR | 20/0.25 | 50 |
| N2 | [128, 64], [128] / 0.3 | 0.0005 | StepLR | 20/0.25 | 80 |
| D1 | [64, 32], [64] / 0.1 | 0.0005 | StepLR | 20/0.25 | 80 |
| D2 | [64, 32], [64] / 0.1 | 0.0005 | StepLR | 20/0.25 | 80 |

In regards to the benchmarking methods, for glasso and nodewise Lasso, we rely on the implementation in `R` package `huge`[18] (Zhao et al., 2012); for RegGMM, we did not find any publicly available packages/modules and thus implemented the method ourselves leveraging `PyTorch`, where $\boldsymbol{\beta}(\boldsymbol{z})$ effectively reduces to a network with a single linear layer.

**Data and code availability.** The resting-state fMRI scans dataset was provided by the authors of Lee et al. (2023). The S&P 100 Index constituents dataset can be collected from Yahoo!Finance[19], with the list of tickers corresponding to the constituents available through Wikipedia[20]. The code repository containing all the implementation is available at `https://github.com/GeorgeMichailidis/covariate-dependent-graphical-model`.

---

[18]`https://cran.r-project.org/package=huge`
[19]`https://finance.yahoo.com`
[20]`https://en.wikipedia.org/wiki/S%26P_100#Components`

