# OpenReview forum: "Covariate-dependent Graphical Model Estimation via Neural Networks with Statistical Guarantees"
_TMLR — Accepted by TMLR_

### Review · Reviewer_FDHz · 2025-02-07

**Summary Of Contributions:**

The paper aims to learn undirected graphical models where the structure is covariate-dependent, and develops an estimation method involving neural network. The method assumes specific function class where the linear coefficients depend (nonlinearly) on some covariate. The authors then provide theoretical guarantee for the finite-sample bounds of the method. Empirical results on synthetic and real data are provided.

**Audience:**

Yes

**Broader Impact Concerns:**

There are no broader impact concerns as far as I am concerned.

**Claims And Evidence:**

Yes

**Requested Changes:**

See above section.

**Strengths And Weaknesses:**

Strengths:
- Extending beyond the Gaussianity assumption is an interesting and important topic in graphical model estimation.
- The estimation algorithm is reasonable.
- Finite samples bounds are provided for the method.
- The empirical results are convincing.

Main comments and/or weaknesses:
- The function class considered seems to be rather restricted.
- What is the benefit/motivation of assuming $f_{jk}(z,x_k)=\beta_{jk}(z)x_k$? Is it precisely about improving statistical efficiency? Can't one just use the general method provided in https://www.jmlr.org/papers/v25/21-0022.html and https://arxiv.org/abs/2305.11379 to handle the more general case of $f_{jk}(z,x_k)$? This should be discussed to make clear the motivation of assuming a more restrictive functional class.
- To make the motivation stronger, it would be better to explain more real-world examples about how the assumed function class will hold, in addition to the example of "age" in the introduction.
- In p. 5, Is the assumption that $x$ and $z$ are bounded in $[-C,C]$ a common assumption? Doesn't that exclude the case where some of the variables are, e.g., Gaussian?
- The rate/bounds in the theorems translate into Euclidean/Frobenius norm (e.g. as mentioned in Remark 6). How does such Frobenius norm consistency translate into bound on graph recovery which often demands more (e.g., see https://proceedings.neurips.cc/paper_files/paper/2008/file/61f2585b0ebcf1f532c4d1ec9a7d51aa-Paper.pdf for the Gaussian case)? If it cannot be translated, then the claim "we establish guarantees on the recovered graph" in the second contribution may not be completely accurate.

---

> ### Author Response · Authors · 2025-03-10
> **RE Q1 & Q2: the function class seems restricted and/or using the general method leveraging the score matrix (PART I)**
>
> Thank you for raising this point. Based on your comment, we have made the following changes to the manuscript (colored in orange):
>
> - Section 1.1: Related Work, we added a paragraph outlining the formulation/approach proposed in [1] and [2] for non-Gaussian data.
> - Section 2: toward the end of the section, we expanded the comment on the exact functional form adopted for $f_{jk}(\boldsymbol{z},x_k)$.
> - Section 6: Discussion, we added a paragraph discussing that the score-based formulation proposed in [1], and briefly contrasting the theoretical result established therein and the one in our manuscript.
>
> Your comment touches upon the following two aspects, discussed in detail next:
> 1. At the formulation level, the score matrix-based approaches akin to [1,2] can be extended to accommodate the additional covariate. However, we opt to adopt a nodewise regression-based formulation, in order to establish finite-sample error bounds under a regime where the number of nodes $p$ can grow slowly with sample size $n$. To contrast, consistency results in [1] are established under an asymptotic regime with $n\rightarrow\infty$, while $p$ remains fixed $p$.
> 2. The exact form of $f_{jk}(\boldsymbol{z},x_k)$ is adopted primarily for interpretability purposes.
>
> **Regarding the more general formulation.** Both [1] and [2] characterize conditional independence in the non-Gaussian setting via the score matrix $\Omega$. Specifically, using the notation consistent with our manuscript, absent of any external covariate $\boldsymbol{z}$, it is defined as
> $$
>     \Omega\_{jk}:= \int |\partial_j\partial_k \log p(\boldsymbol{x})|^2 p(\boldsymbol{x})\mathrm{d}\boldsymbol{x} = \mathbb{E}\_{p(\boldsymbol{x})} \big[ \partial_j\partial_k \log p(\boldsymbol{x})  \big]^2.
> $$
> To estimate $\Omega$, [1] proceed with approximating the target density via a triangular transport map;  [2] consider an implicit score matching objective function and additionally incorporate a sparsity-inducing penalty term. In the presence of an external covariate, at the formulation level,
> with $x_j\perp x_k|(\boldsymbol{x}\_{-jk},\boldsymbol{z})$, the joint-on-$\boldsymbol{x}$-conditional-on-$\boldsymbol{z}$ density factorizes as
> $p(\boldsymbol{x}| \boldsymbol{z}) = p(x_j| \boldsymbol{x}\_{-jk}, \boldsymbol{z})p(x_k| \boldsymbol{x}\_{-jk}, \boldsymbol{z})p(\boldsymbol{x}\_{-jk}, \boldsymbol{z})$.
> Under the assumption that $p(\boldsymbol{x}|\boldsymbol{z})$ is continuously differentiable, the score matrix can be analogously written as
> $$
>     \Omega_{jk}(\boldsymbol{z}):= \int |\partial_j\partial_k \log p(\boldsymbol{x}|\boldsymbol{z})|^2 p(\boldsymbol{x}|\boldsymbol{z})\mathrm{d}\boldsymbol{x} = \mathbb{E}_{p(\boldsymbol{x}|\boldsymbol{z})} \big[ \partial_j\partial_k \log p(\boldsymbol{x}|\boldsymbol{z})  \big]^2.
> $$
> Transport map-based and score matching-based estimation can respectively be carried out by referencing the procedure outlined in [3] and [4], where the conditional density/score functions are estimated.
>
> The theoretical result established in [1] is under an _asymptotic_ regime with a fixed number of nodes $p$ and increasing sample size $n$. Specifically, it focuses on the _asymptotic_ recovery of the edge set under thresholding: Proposition 9 establishes that when the thresholding function $f(n)$ grows with $n$ at an appropriate rate, the estimated edge set after thresholding is asymptotically correct. The analysis relies on somewhat strong assumptions that (1) the transport map parametrization---through finite basis expansion---is sufficiently rich to recover the target density (which can be made analogous to a well-specified setting under the PAC-learning framework), and (2) the estimated $\Omega$ before thresholding is given by the exact expectation of the functional of the estimated density, rather than the sample-level estimate. Under this "fixed-$p$" asymptotic regime, the authors proceed with the analysis leveraging Taylor expansion; subsequent steps are facilitated by the aforementioned assumptions, wherein delta method and the continuous mapping theorem are invoked.
>
> On the other hand, we are interested in establishing a finite-sample error bound for the estimator under a "growing-$p$" (albeit slowly) regime, which requires a different set of technical tools. In particular, by considering a supervised learning formulation, we are able to leverage existing results for ERM and MLP-style neural networks, and establish the finite-sample error bounds under moderate assumptions. In fact, it is unclear how bounds of similar type can be established for the estimators considered in [1,2] in the absence of the covariate $\boldsymbol{z}$, whose presence can add further complications to the issue.
>
> As such, instead of adopting an approach based on the score matrix proposed in [1], we consider a node-wise regression formulation, which allows us to leverage existing results in the supervised learning literature.

---

> > ### Author Response · Authors · 2025-03-10
> > **RE Q5: how Frobenius norm consistency translates into the bound on graph recovery (PART I)**
> >
> > Thank you for raising this interesting question, which prompted us to consider this issue more deeply.
> >
> > We agree that in high-dimensional settings---in particular, linear models with sparse parameters---one can translate Frobenius/$\ell_2$ norm consistency of the model parameter to support recovery with some extra assumptions,
> > such as minimum signal strength of the parameter ("$\beta$-min" condition) and incoherence conditions on the design matrix [1,2],
> > wherein the roadmap to establish such results follows a primal-dual witness strategy. However, for the setup considered in this paper, trying to establish similar results is challenging and not straightforward.
> > Next, we elaborate using a bare-bones regression model in the form of $y=\sum_{k=1}^p\beta_k(\boldsymbol{z})x_k + \varepsilon$ under the ERM framework; $\beta_k(\boldsymbol{z}):\mathbb{R}^q\mapsto \mathbb{R}$'s are parameterized by neural networks as estimation proceeds.
> > Note that for the notion of support recovery to be meaningful, the underlying true data generating mechanism needs to satisfy $\beta_k^*(\boldsymbol{z})\equiv 0$ for all $\boldsymbol{z}$, whenever $k\in\mathcal{S}^c$, the complement of the support set;
> > in other words, the true model becomes $y=\sum_{k\in\mathcal{S}} \beta^*_k(\boldsymbol{z})x_k + \varepsilon$, with $\mathcal{S}$ denoting the support set of the "active" predictors.
> >
> >
> > Leveraging an analogous strategy to establishing support recovery in $\ell_1$-penalized sparse linear regression,
> > one proceeds with expanding $\beta_k(\boldsymbol{z})$ onto the appropriate basis functions, so that an equivalent notion of a "design" matrix can be considered; for example, in the case where $\beta_k(\boldsymbol{z})$ is parameterized by a one-layer MLP with $H$ hidden neurons and ReLU activation, namely $\beta_k(\boldsymbol{z}) = (\boldsymbol{w}^{(k,2)})^\top \sigma(W^{(k,1)}\boldsymbol{z} + \boldsymbol{b}^{(k,1)}) = \sum_{h=1}^H w^{(k,2)}\_h \sigma(W^{(k,1)}\_{h,:}\boldsymbol{z} + b^{(k,1)}\_h)$, one can view $\phi^h_k(\boldsymbol{z}):=\sigma(W^{(k,1)}_{h,:}\boldsymbol{z} + b^{(k,1)}_h)$ as the basis. This gives rise to a design matrix, whose columns are sample versions of $\phi_k^h(\boldsymbol{z})x_k; k=1,\cdots,K; h=1,\cdots,H$. However, the challenges are two-fold for the ensuing technical developments:
> >
> > - Although an incoherence-type condition can be imposed on such a design matrix, these bases are not "fixed", but depend on learnable parameters.
> > This is in contrast to previous work in sparse additive model, where the functions $\beta_k(\boldsymbol{z})$ are expanded in, e.g.,
> > B-spline bases [3] and the required incoherence condition is imposed on the "design" matrix induced by these fixed bases.
> > Moreover, by expanding on a fixed set of basis functions, it is sufficient to estimate their corresponding coefficients and enforce sparsity
> > on these coefficients; consequently, $\ell_1$-type penalties are sufficient. On the other hand, in our setting,
> > a more involved penalty that controls in addition the "magnitude" of the function itself may be required (see, e.g., [4]).
> > - The fact that the functional class of the working model (e.g., parameterizing $\beta_k(\cdot)$ by an MLP) is only an approximation to the underlying true data generating mechanism---namely, the setting is mis-specified and hence there is approximation error---can induce further complications.
> >
> > To summarize, establishing consistency results for graph support recovery under the ERM framework is arguably a separate topic on its own. Theoretical results even for a simpler regression setting along this direction remain largely unexplored, due to the fact that the technical challenges outlined above need to be carefully addressed and resolved.

---

> ### Author Response · Authors · 2025-03-10
> **RE Q1 & Q2: the function class seems restricted and/or using the general method leveraging the score matrix (PART II)**
>
> **Regarding the specific functional form adopted.** In regards to the restricted form adopted for $f_{jk}(\boldsymbol{z},x_k)\equiv\beta_{jk}(\boldsymbol{z}) x_k$, the consideration is mainly due to interpretability considerations.
> First, note that without a separable structure between $\boldsymbol{z}$ and $x_k$, namely, when $f_{jk}(\boldsymbol{z},x_k)$ allows for general interaction between $\boldsymbol{z}$ and $x_k$, one cannot readily extract the conditional independence graph (weighted). Concretely, in the case where $f_{jk}(\boldsymbol{z},x_k)\equiv 0$, indeed one can infer $x_j\perp x_k\,|\,(\boldsymbol{x}\_{-jk},\boldsymbol{z})$;
> however, when $f_{jk}(\boldsymbol{z},x_k)\neq 0$, to determine the strength of the dependency, one needs to obtain it post-hoc, e.g., resorting to quantities in a form similar to $\nabla_{x_k}f_{jk}(\boldsymbol{z},x_k)$. On the other hand, with a separable structure between $\boldsymbol{z}$ and $x_k$ and thus $f_{jk}(\boldsymbol{z},x_k)\equiv\beta_{jk}(\boldsymbol{z})\gamma_{jk}(x_k)$, it behaves like a "locally linear" model on $\gamma_{jk}(x_k)$; the edge weight can be summarized via $\beta_{jk}(\boldsymbol{z})$, which can be interpreted as the strength of relevance (see also [5]) and is a "primary" quantity that can be directly estimated. Finally, $\gamma_{jk}(x_k)\equiv x_k$ is a practical simplification. One can opt for a more complex $\gamma_{jk}(x_k)$ and learn it end-to-end together with the $\beta_{jk}$'s, although for graphical model estimation problem in question, the "raw features" (namely the $x_k$'s) are meaningful themselves, unlike, for example, pixels in an image (see also Section 4 in [5]). Performance-wise, we have experimented with the case of incorporating $\gamma_{jk}(x_k)$ parameterized by neural networks; for the non-Gaussian settings considered in the manuscript, the improvement in performance is minimal.
>
>
> In summary, a formulation that arguably is more general can be pursued by extending [1,2] (to incorporate the extra dependency on the covariate $\boldsymbol{z}$), whose estimation can proceed with estimating the corresponding conditional density or score function; yet, the roadmap for establishing finite-sample theoretical properties of estimators along this line remains unclear. The formulation we adopted promotes interpretability, and it allows us to establish finite-sample error bounds for the estimated quantities, leveraging existing tools in the literature.
>
>
> References
>
> [1] Baptista, R., Morrison, R., Zahm, O., & Marzouk, Y. (2024). Learning non-Gaussian graphical models via Hessian scores and triangular transport. _Journal of Machine Learning Research_.
>
> [2] Zheng, Y., Ng, I., Fan, Y., & Zhang, K. (2023). Generalized precision matrix for scalable estimation of nonparametric markov networks. _ICLR_.
>
> [3] Baptista, R., Marzouk, Y., & Zahm, O. (2024). On the representation and learning of monotone triangular transport maps. _Foundations of Computational Mathematics_.
>
> [4] Dasgupta, A., Murgoitio-Esandi, J., Ray, D., & Oberai, A. (2023). Conditional score-based generative models for solving physics-based inverse problems. _NeurIPS Workshop on Deep Learning and Inverse Problems_.
>
> [5] Alvarez Melis, D., & Jaakkola, T. (2018). Towards robust interpretability with self-explaining neural networks. _NeurIPS_

---

> ### Author Response · Authors · 2025-03-10
> **RE Q3 (more real-word examples) & Q4 (boundedness)**
>
> **RE Q3: more real-world examples in additional to the example of "age" in the introduction**
>
> In the revised manuscript, we added an example (in Section 1) on gene co-expression networks, wherein the SNPs (potentially a large-dimensional vector) are covariates that impact the connectivity.
>
> **RE Q4: the assumption that $\boldsymbol{x}$ and $\boldsymbol{z}$ are bounded in $[-C,C]$**
>
> Under the PAC learning framework, the boundedness assumption is a standard one, due to the fact that it enables the application of  Talagrand's concentration inequality used in the Local Rademacher complexity analysis [1]. The extension to unbounded settings is non-trivial; for example, [2] elaborates why the key result established in [1] would fail in the presence of Gaussian and/or heavy-tailed noise, or in unbounded regression settings.  Arguably, relaxing the boundedness assumption in even simpler setups---e.g., standard regression settings---under the PAC learning framework is a line of ongoing research on its own.
>
> References
>
> [1] Bartlett, P. L., Bousquet, O., & Mendelson, S. (2005). Local rademacher complexities. _Annals of Statistics_
>
> [2] Mendelson, S. (2014). Learning without concentration. _COLT_

---

> ### Author Response · Authors · 2025-03-10
> **RE Q5: how Frobenius norm consistency translates into the bound on graph recovery (PART II)**
>
> **Additional results for edge-wise recovery in the revised manuscript.**
> Based on your comments, we incorporated Corollary 2 that provides an error bound for individual edges (in expectation),
> which refines the consistency results in Corollary 1 under an additional "minimum eigenvalue bounded away from zero" type condition.
> The setting considered therein is that the edges can be segmented into strong and weak ones, and there is some
> "separation" between the two in terms of their magnitude. These results are in Section 3, highlighted in orange,
> whose proof is deferred to Appendix A.7 (p.25). Finally, in Section 1, we have also revised our theoretical contribution,
> which now reads as "On the theoretical front, we establish guarantees on the recovered _edges_ based on a deep neural network estimator."
>
>
>
> References
>
> [1] Raskutti, G., Yu, B., Wainwright, M. J., & Ravikumar, P. (2008). Model selection in Gaussian graphical models: high-dimensional consistency of $\ell_1 $-regularized MLE. _NeurIPS_
>
> [2] Wainwright, M. J. (2009). Sharp thresholds for High-Dimensional and noisy sparsity recovery using $\ell_1$-constrained quadratic programming (Lasso). _IEEE Transactions on Information Theory_
>
> [3] Meier, L., Van de Geer, S., & Bühlmann, P. (2009). High-dimensional additive modeling. _Annals of Statistics_
>
> [4] Tan, Z., & Zhang, C. H. (2019). Doubly penalized estimation in additive regression with high-dimensional data. _Annals of Statistics_

---

> > ### Comment · Reviewer_FDHz · 2025-04-06
> >
> > Thanks for the detailed response. My concerns have been addressed, and I have updated my review accordingly.

---

### Review · Reviewer_Ppck · 2025-02-16

**Summary Of Contributions:**

This paper studies the following problem.
Suppose $G$ is a graph with $p$ vertices.
Let $x$ be the $p$-dimensional random vector where each entry corresponds to a vertex.
We consider the following model that $x$ satisfies
$$
x_j = \sum_{k\neq j} \beta_{jk}(z) x_k + \epsilon_j
$$
where $\beta_{jk}(z) \neq 0$ if there is an edge between the vertex $j$ and $k$, $\epsilon_j$ is an independent random variable whose expectation is $0$ and $z$ is a latent variable.
We further assume that $\beta_{jk}(z)$ is parametrized by a neural network and the neural network is parametrized by $\theta$.
Namely, we rewrite $\beta_{jk}(z)$ to be $\beta_{jk}(z,\theta)$.

Now, suppose we are given $n$ samples $\{(x^i,z^i)\}$ generated by the aforementioned model.
The authors designed an algorithm to approximate the neural network parameter $\theta$ and hence $\beta_{jk}$.
The main result of this paper is to show that the generalization error decays in the rate of $\frac{1}{n}$ (up to log factors).
The authors also provide the experimental results.

**Audience:**

Yes

**Broader Impact Concerns:**

The work is mainly a theoretical result.

**Claims And Evidence:**

Yes

**Requested Changes:**

.

**Strengths And Weaknesses:**

Strengths:

- This paper provides an interesting theoretical result.
In particular, I have never seen the usage of Dudley's integral in this context.

- The presentation is generally clear.

Weaknesses:

- Though the authors provide experimental results to support their theoretical guarantees, the model considered in the paper may be oversimplified.

---

> ### Author Response · Authors · 2025-03-10
> **RE: the simplified model specification**
>
> Thank you for your review of our work and the comments! We acknowledge that the model considered is somewhat simplified.
> There are two aspects to it:
> - we adopt a node-conditional formulation, which allows us to establish finite-sample error bound for the estimator, leveraging tools developed under the ERM framework;
> - the specific functional form adopted is primarily out of interpretability consideration.
>
> They are briefly illustrated next.
>
> **At the formulation level**, a potentially more general one is to extend approaches that leverage the score matrix proposed in [1].
> Specifically, by characterizing the conditional dependence for non-Gaussian data via $\Omega\_{jk}:= \mathbb{E}\_{p(\boldsymbol{x})} \big[ \partial_j\partial_k \log p(\boldsymbol{x})\big]^2$,
> estimation proceeds with estimating $p(\boldsymbol{x})$ through a lower triangular transport map [1], or approximating $\nabla\_{\boldsymbol{x}} \log p(\boldsymbol{x})$ by score matching [2].
> This line of work can be extended to incorporate covariates $\boldsymbol{z}$: the score matrix is modified to
> $$
>     \Omega\_{jk}(\boldsymbol{z}):= \int |\partial_j\partial_k \log p(\boldsymbol{x}|\boldsymbol{z})|^2 p(\boldsymbol{x}|\boldsymbol{z})\mathrm{d}\boldsymbol{x} = \mathbb{E}\_{p(\boldsymbol{x}|\boldsymbol{z})} \big[ \partial_j\partial_k \log p(\boldsymbol{x}|\boldsymbol{z})  \big]^2,
> $$
> and one proceeds with estimating  $p(\boldsymbol{x}|\boldsymbol{z})$ or $\nabla_{\boldsymbol{x}} \log p(\boldsymbol{x}|\boldsymbol{z})$.
> Theoretical results are provided in [1], however, under a fixed-$p$-growing-$n$ asymptotic regime. In particular, the consistency result is built on a Taylor expansion,
> followed by steps that require somewhat strong assumptions to invoke the delta method and the continuous mapping theorem.
> We are interested in a regime where the number of nodes $p$ can grow slowly with sample size $n$; the result we established is the finite-sample error bound of the estimator which requires a different set of technical tools from those under the fixed-$p$ regime.
> To this end, we consider a node-conditional formulation, which reduces the learning problem to an ERM one, enabling us to leverage existing tools.
>
> **In regards to the specific functional form adopted**, it is largely out of interpretability considerations. The core structural assumption embedded in our formulation is a separable structure between $\boldsymbol{z}$ and $\boldsymbol{x}\_k$
> in the additive component functions, namely, $x_j = \sum_{k\neq j} f_{jk}(\boldsymbol{z}, x_k) + \varepsilon_j$ with $f_{jk}(\boldsymbol{z}, x_k)\equiv \beta_{jk}(\boldsymbol{z})\gamma_{jk}(x_k)$,
> which allows us to extract the weighted edge that measures the strength of the dependency between $x_j$ and $x_k$ directly through $\beta_{jk}(\boldsymbol{z})$. For more general interaction in the generic form of $f_{jk}(\boldsymbol{z}, x_k)$,
> one would need to resort to post-hoc operations to quantify the magnitude of dependency, e.g., via $\nabla_{x_k}f_{jk}(\boldsymbol{z}, x_k)$. Finally, for $\gamma_{jk}(x_k)\equiv x_k$, it is a practical simplification w.r.t. any more complex $\gamma_{jk}(x_k)$ that can be learned
> end-to-end together with the $\beta_{jk}$'s.
>
> In the revised manuscript, these two aspects elaborated in detail (changes colored in orange):
> - Section 2: we expanded the explanations for why the exact functional form for $f_{jk}(\boldsymbol{z}, x_k)$ is adopted;
> - Section 6: we commented on a potentially more general formulation at the methodological level, which extends existing work for non-Gaussian data absent of the covariates (outlined in Section 1.1); we also contrasted the type of theoretical results established therein to ours.
>
> References
>
> [1] Baptista, R., Morrison, R., Zahm, O., & Marzouk, Y. (2024). Learning non-Gaussian graphical models via Hessian scores and triangular transport. _Journal of Machine Learning Research_.
>
> [2] Zheng, Y., Ng, I., Fan, Y., & Zhang, K. (2023). Generalized precision matrix for scalable estimation of nonparametric markov networks. _ICLR_.

---

### Review · Reviewer_e16v · 2025-03-15

**Summary Of Contributions:**

The submission introduces a deep neural network-based methodology for estimating covariate-dependent graphical models, where the structure of a graphical model (representing conditional independencies) can vary continuously with external covariates. The paper provides finite-sample statistical guarantees (PAC-learning type bounds), ensuring theoretical rigor and convergence rates under stated assumptions. Empirically, it demonstrates superior performance over existing methods in both synthetic scenarios and real-world datasets (neuroscience and finance).

**Audience:**

Yes

**Claims And Evidence:**

Yes

**Requested Changes:**

Despite the overall clarity, there are a few areas that could be improved or elaborated:

1. Symmetry of the Graph: The model (1) treats each node’s neighborhood separately, and the neural network outputs $\beta_{jk}(z)$ potentially for each ordered pair. It’s implied that in an undirected graph, $\beta_{jk}(z)$ and $\beta_{kj}(z)$ should both be zero or non-zero when an edge is absent or present. Unless I missed it, the authors did not explicitly state how they enforce or utilize this symmetry. In practice, they might take the union of edges predicted in each direction or average the two estimates. A brief note on this (e.g., “we treat an edge $(j,k)$ as present if either $\beta_{jk}$ or $\beta_{kj}$ exceeds the threshold” or similar) would clarify the estimation of an undirected skeleton from directed regression outputs. This is a minor point, as it’s understood from context and common practice in neighborhood selection, but mentioning it would remove any ambiguity.

2. Regularization and Tuning: I don't think there is an explicit mention about the form of the regularizer used, if any, in the main text. I suspect the authors likely used an $\ell_1$ penalty on the coefficients to encourage many $\beta_{jk}(z)$ to be exactly zero. An equation for the full objective (loss + penalty) would be helpful. Also, while the experiments indicate the authors thresholded the learned weights at a certain level, it’s not fully explained how that threshold was chosen. Did they use a validation set to pick an optimal threshold or lambda for the penalty? Or did they simply examine a range of thresholds (as in Table 3) to show stability? A sentence on the model selection procedure (e.g., using cross-validation for any hyperparameters, or using BIC to select a sparsity level) would strengthen the presentation.

3. Summarize briefly differences and similarities with prior approaches (linear-varying, kernel-based, and score-matching methods) in a concise comparative table in the related-work section. This would facilitate easier comparison and highlight clearly the novelty and advantages of the proposed approach.

**Strengths And Weaknesses:**

### Strong aspects:
- Theoretical derivations, assumptions, and results (PAC-style guarantees and finite-sample error bounds) are rigorous and seem technically sound.
- Rigorously combine deep neural network-based functional approximation with covariate-dependent graphical models, providing a clear theoretical foundation.
- Reasonable experimentation with appropriate baselines (linear and static graphical models), clear illustration of scenarios where the proposed approach excels (particularly nonlinear dependencies).
- Does not require Gaussianity; accommodates general non-Gaussian distributions using node-wise regression and score matching methods, substantially broadening applicability.
- Clear real-world examples demonstrating interpretability, such as cognitive-state-dependent brain networks and finance market volatility-dependent stock correlations.

### Weaker elements (minor, but worth attention):
- The process of translating continuous estimated weights to a symmetric undirected graph is not fully clear.
- Not clear if regularization was used in the experiments and what are the parameter tuning strategies (e.g., choice of sparsity penalty or threshold).

---

> ### Author Response · Authors · 2025-03-21
> **Response to Reviewer e16V**
>
> Thank you for your careful review of our work and your comments. We incorporated your constructive suggestions and made the following changes (highlighted in orange) to the manuscript:
> * Added Table 1 (Section 1.1 Related Work, p.3) that summarizes existing work on covariate-dependent graphical model estimation.
> * Added Remark 2 (end of Section 2, p.6), which discusses how to obtain a sparsified graph via thresholding and "symmetrize" the skeleton.
> * Added a sentence in Appendix E, stating that the hyperparameters are chosen based on the validation set performance.
>
> Please find our point-by-point responses below.
>
> **RE: a brief note on ensuring the "symmetry" of the undirected skeleton**
>
> Thank you for the suggestion. In the revised manuscript, we added Remark 3, stating that to obtain a sparsified graph, we consider a thresholding procedure and adopt the "AND" rule (i.e., both $\widehat{\beta}\_{jk}$ and $\widehat{\beta}\_{kj}$ being nonzero after thresholding) to determine the existence of an undirected edge.
>
> **RE: regularization and hyperparameter tuning**
>
> As briefly noted in the newly added Remark 3 (as well as the results on "Extensions to edge-wise recovery" in Section 3), we do not posit the assumption that the underlying true graph is (exactly) sparse per se; as such, no sparsity-inducing regularization term is incorporated in the objective function. In practice, we resort to a post-hoc thresholding procedure to sparsify the graph for interpretability purposes, and establish theoretical results under scenarios where the edges can be separated into weak and strong ones based on their magnitude. It is worth noting though, that given the "functional" nature of the estimated $\beta_{jk}$'s, analyzing its theoretical properties in the presence of a regularization term (e.g., $\ell_1$ penalty) can be fairly involved; see, e.g., [1].
>
> In regards to hyperparameter tuning, in the revised manuscript, we added a sentence in the existing Appendix E, noting that the optimal set is selected based on the performance on a validation set, with the MSE from nodewise regression as the criterion.
>
> **RE: a table summarizing prior approaches**
>
> Thank you for this suggestion; Table 1 has been added in Section 1.1, which focuses on outlining existing work on covariate-dependent graphical models. Given that the existing work using a score-matrix based formulation is under a setting _without_ covariate dependency, we opt not to include it in the table and only mention the possibility of extensions based on it; see the newly added sentence (in orange) above Table 1.
>
> [1] Tan, Z., & Zhang, C. H. (2019). Doubly penalized estimation in additive regression with high-dimensional data. _Annals of Statistics_

---

### Author Response · Authors · 2025-04-24
**Acknowledgements to the Action Editor and Reviewers; camera-ready revision uploaded.**

We sincerely thank the Action Editor and three reviewers for their careful review of the manuscript, and for their constructive comments and suggestions, which helped improve the quality of the paper.

We have uploaded the camera-ready version of the manuscript, and included the link to the code repository.

---

### Decision · Action_Editor_pjXs · 2025-04-19

**Recommendation:** Accept as is

**Comment:**

The authors have addressed all reviewers' concerns.

**Audience:**

All reviewers agree that the manuscript has an audience. Extensions to the classical graphical model estimation problem as well as novel algorithms with guarantees are definitely relevant to the probabilistic graphical model subcommunity and the TMLR readership. The audience might be relatively smaller as compared to the whole machine learning community, but it is important to let the ML community know of these developments, and to increase the diversity of topics being studied.

**Claims And Evidence:**

All reviewers agree that the manuscript presents appropriate evidence for its claims. The manuscript proposes an algorithm to estimate a covariate-dependent graphical model, where there is an external variable modulating the weights of the node-conditional distributions. Generalization bounds are provided. Experimental evidence is also provided on synthetic and real world data.